# Learning and Reusing Abstract Latent Actions in a Hippocampal-Entorhinal-Inspired World Model

## Abstract

Humans are capable of abstracting dynamic experiences into structured representations, facilitating both the inference of shared patterns by observing similar transition dynamics and the transfer of these structures across varied contexts. The hippocampal-entorhinal circuit, widely known for its role in spatial navigation, also supports the representation of abstract conceptual spaces crucial for non-spatial cognitive processes. This function emerges from the distinct yet integrated encoding of content-specific details by the hippocampus and abstract structures by the entorhinal cortex, facilitating structural generalization across varied contexts. Although the hippocampal-entorhinal circuit has been previously explored as a predictive system for binding contents, the process for concurrently extracting abstract structures from continuous real-world dynamics remains largely understudied. In this work, we propose a computational model inspired by the hippocampal-entorhinal circuit, capable of simultaneously inferring latent actions to form abstract structures and constructing predictive world models from real-world video sequences. Our model combines an inverse model for extracting abstract latent actions with a hippocampal-entorhinal-inspired coupling model that separately encodes contents and structures, leveraging action-driven path integration for prediction. Experimental results demonstrate that our model effectively captures abstract latent actions, reuses them robustly across diverse contexts, and achieves reliable predictive performance in both familiar and novel environments. Additionally, our analysis of latent representations from 3D object rotation datasets highlights why latent actions extracted through entorhinal cortex representations demonstrate greater abstraction and reusability. This work provides novel insights into the brain-inspired mechanisms underlying the self-supervised learning of abstract latent actions and world models from real-world dynamics, illuminating cognitive processes essential for transfer learning and data-efficient learning.

## 1 Introduction

The hippocampal-entorhinal circuit has traditionally been studied in the context of spatial memory and navigation (Keefe & Nadel, 1978; Hafting et al., 2005). Recently emerging research demonstrates that this circuit extends beyond physical space navigation to encode abstract conceptual spaces, supporting diverse high-level cognitive tasks (Behrens et al., 2018). This neural architecture provides cognitive scaffolds for understanding abstract relationships by factorizing content and structure representations (Lerousseau & Summerfield, 2024). Substantial experimental and theoretical evidence (Manns & Eichenbaum, 2006; Whittington et al., 2020) supports a functional division: the hippocampus (HPC) binds content-specific information from individual experiences (Eichenbaum, 2017), while the medial entorhinal cortex (MEC) encodes abstract structures (Julian et al., 2018; Bao et al., 2019). This separation enables structural generalization, allowing the system to bind extracted structural representations flexibly with novel contexts (Kemp & Tenenbaum, 2008).

Grid cells within the MEC are fundamental to constructing these abstract structures (Behrens et al., 2018). Characterized by their periodic hexagonal firing patterns at various spatial scales (Giocomo et al., 2011), grid cells with the same spacing are organized into modules that function as continuous attractor neural networks (CANNs) (Amari, 1977; Ben-Yishai et al., 1995; Wu et al., 2008).

Furthermore, when receiving velocity inputs, grid cells drive network activity across this attractor manifold, enabling path integration in abstract spaces (Burak & Fiete, 2009; Gardner et al., 2022) and facilitating mental simulation and planning.

The coupling between the HPC and MEC enables the binding of specific context information to abstract structures (Whittington et al., 2020; Chandra et al., 2025; George et al., 2021). This circuit predicts subsequent states by performing path integration within the MEC, generating predictions specific to environmental contexts. This function directly parallels the concept of "world model" (Ha & Schmidhuber, 2018; LeCun, 2022), suggesting that the hippocampal-entorhinal circuit serves as a biological implementation of a world model. In addition, the abstract structures in the MEC parallel the notion of latent actions in policy learning (Bruce et al., 2024; Schmidt & Jiang, 2024), where latent actions serve as compact representations of action transition dynamics, such that similar transitions yield similar latent codes. These correspondences inspire us to develop a brain-inspired model that generalizes to real-world scenarios. It abstracts the principle functions of the HPC-MEC circuits and takes the form of a disentangled world model which learns to extract complex transition dynamics.

Our research addresses two fundamental questions: How does a model simultaneously learn latent representations of concrete contents and extract abstract, reusable latent actions from concrete sequences without prior knowledge of actions, and how can these abstract structures be leveraged to facilitate generalization across diverse objects and environments?

In this work, we propose a disentangled world model inspired by the hippocampal-entorhinal circuit capable of concurrently inferring abstract latent actions and learning a meaningful latent space from real-world sequences. The model comprises two components: an inverse model that extracts abstract latent actions from sequences (Section 3.2), and an HPC-MEC-inspired world model that disentangles abstract structures from contents while learning to predict the next frame through action-driven path integration (Section 3.1). Our model demonstrates robust capabilities of flexibly reusing latent actions across different environments and objects. Furthermore, it exhibits effective predictive performance in real-world human activity scenarios and generalizes to previously unseen environments (Section 4).

The key contributions of this work are as follows:

- **Self-supervised learning of abstract latent actions and HPC-MEC world model:** We introduce a self-supervised framework that jointly infers abstract latent actions and learns a HPC-MEC-inspired world model. The input consists solely of observation sequences, without requiring explicit prior on abstract structures or ground-truth action labels.

- **Learning and reusing abstract latent actions across different contexts:** We propose a brain-inspired model that learn to extract shared structures from similar transition dynamics and flexibly reuses abstract latent actions across varied environments and object categories. This transfer capacity is enabled by the integration of abstract structures encoded in the MEC with content-specific details stored in the HPC. Our model also shows generalization capabilities to out-of-distribution datasets.

- **Analysis of learned latent spaces using rotational dynamics:** We use rotational dynamics as a controllable example to show that the model successfully decouples appearance from dynamics, demonstrating periodicity and in-class structure sharing.

## 2 RELATED WORK

**Cognitive map models.** Cognitive map models typically attribute structural abstraction to MEC, sensory binding to HPC, and sensory prediction to their interaction. The Tolman-Eichenbaum Machine (TEM) (Whittington et al., 2020) extends beyond spatial domains by using recurrent networks to form cognitive maps that predict observations from predefined actions, though it requires re-learning abstract maps across environments. Clone-structured cognitive graphs (CSCG) (George et al., 2021) offer graph-based Markovian representations of structural relationships without prior constraints, but remain limited to discrete domains. Vector-HaSH (Chandra et al., 2025) generates velocity inputs from hippocampal states to drive grid cells forming episodic memories, but its velocity vectors are learned by memorizing the whole sequence. All these works lack the critical step of inferring shared abstract structure from sequences in continuous and real-world environments.

**Abstract velocity extraction.** In neuroscience, Iyer et al. (2024) employs an inverse model to extract abstract velocities from high-dimensional observations and maps them to low-dimensional grid cell velocity inputs. However, this approach significantly simplifies the complex representation learning in HPC-MEC circuits, focusing only on simple artificial stimuli.

**Infer latent actions from the observations.** World models (Ha & Schmidhuber, 2018; LeCun, 2022) are generative models that predict future observations based on past observations and actions. When ground-truth actions are unavailable, inverse and forward models are commonly employed to infer actions and predict future states from observation-only demonstrations (Bruce et al., 2024; Ye et al., 2022; Schmidt & Jiang, 2024). Genie (Bruce et al., 2024), FICC (Ye et al., 2022), and LAPO (Schmidt & Jiang, 2024) primarily target 2D gaming environments, while LAPA (Ye et al., 2024), Moto (Chen et al., 2024b), IGOR (Chen et al., 2024a), and UniVLA (Bu et al., 2025) focus on latent action extraction from real-world settings for pretraining policy models. Although LAPA, Moto, IGOR, and UniVLA extract latent actions using VQ-VAE objectives, they are not explicitly designed for interpretable state-action interaction and focus primarily on utilizing latent actions as surrogate actions. Our approach implements state-action interaction through grid cell path integration, ensuring that latent actions function explicitly as velocities acting on current states. AdaWorld (Gao et al., 2025) achieves latent action transfer but relies on a large-scale pretrained video diffusion model, which is not the mechanism by which the HPC-MEC circuit accomplishes action transfer.

## 3 Self-supervised learning of latent actions and a world model

The model is mainly separated into two parts: the HPC-MEC coupling model (Fig. 1(A, B)) and the inverse model (Fig. 1(C)). First, we use the pretrained multi-scale VQ-VAE (Tian et al., 2024) to extract observation embeddings from video sequences. The HPC-MEC coupling model encodes contents and structures separately and predicts the next frame using action-driven path integration (Fig. 1(B)). The inverse model is used to infer latent actions from the MEC embeddings. Finally, the decoder of VAE reconstructs the next frame from the generated observation embedding. The overall process is illustrated in Fig. 1.

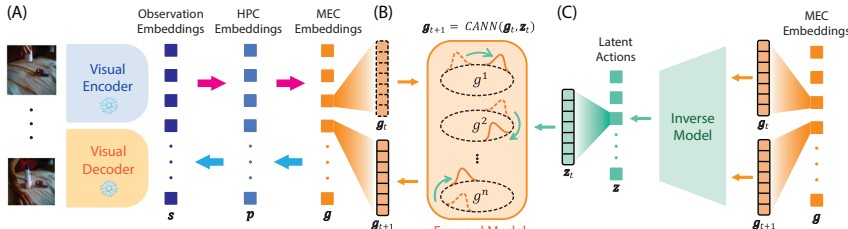

Figure 1: **Overview of the model architecture.** (A) Video clips are passed through the visual encoder to obtain observation embeddings $s$, which are encoded through the HPC to produce HPC embeddings $p$, and then passed to the MEC to generate MEC embeddings $g$. Finally, the generative pathway decodes them into the observation. The multi-scale VAE is fixed during training. (B) The latent action $z_t$ operates on the MEC embedding $g_t$ at time $t$ using continuous attractor dynamics to generate the next MEC embedding $g_{t+1}$. (C) The inverse model is used to infer latent actions from the MEC embeddings $g$.

### 3.1 The HPC-MEC coupling model serves as a world model

The HPC-MEC coupling model is a hierarchical encoder-decoder architecture comprising two principal pathways: the visual inference pathway and the generative pathway. Rather than functioning solely as a reconstruction-based encoder–decoder, the model performs path integration by applying latent actions to its MEC embeddings, enabling it to generate subsequent observations and thus serves as a world model. The graphical model of the HPC-MEC coupling model is illustrated in Fig. 2(A).

**The visual inference pathway.** Input video frames $o_{1:T}$ are processed by the VAE encoder to obtain observation embeddings $s_{1:T}^{\text{inf}}$ (see Appendix B.3 for details). $s_{1:T}^{\text{inf}}$ are then encoded into higher-dimensional HPC embeddings $p_{1:T}^{\text{inf}}$ that capture the content information. Finally, the MEC compresses $p_{1:T}^{\text{inf}}$ into lower-dimensional MEC embeddings $g_{1:T}^{\text{inf}}$. Both the HPC and MEC use

spatial-temporal Transformer encoders with temporal causal masking (Ye et al., 2024) to capture time dependencies.

**The generation pathway.** The transition dynamic of the MEC is implemented as CANN-inspired template matching (Wu et al., 2008; Yoon et al., 2013) capable of performing path integration. A Continuous Attractor Neural Network (CANN) is a specific type of recurrent neural network designed to maintain a stable, continuous representation of information in its activity pattern. The CANN maintains the current structural state as a stable, localized "bump" of neural activity within its metric space. This inherent geometric regularity is what makes them well-suited for path integration of grid cells (Burak & Fiete, 2009; Gardner et al., 2022). The key is that the network's translation-invariant geometric structure facilitates flexible state transitions through operators (see Appendix B.4 for mathematical details). In our model, the inferred latent action $z_t$ serves as precisely such an operator. The latent action controllably shifts the activity bump along the network's metric axes. To simplify the CANN dynamics, each dimension of the MEC embedding $g_t$ is represented by the bump center of a one-dimensional CANN: $g_t = [g_t^1, g_t^2, \ldots, g_t^n]$, where $n$ is the number of CANNs and $g_t^i$ is the bump center of the $i$-th CANN at time $t$ (Fig. 2(B)). Then $z_t$ is transformed into a concrete velocity term through its integration with the MEC embedding $g_t$. Specifically, the model first maps the concatenation of $z_t$ and $g_t$ to produce a displacement vector $\Delta g_t$, which serves as a direct velocity input to the CANN dynamics. The path integration dynamic can be simplified as the following equation at one discrete time step:

$$g_{t+1}^{\text{gen}} = g_t^{\text{gen}} + \Delta g_t = g_t^{\text{gen}} + f_{\text{forward}}(z_t, g_t^{\text{gen}}), \tag{1}$$

where $f_{\text{forward}}$ is implemented as a MLP. This update rule allows each dimension of $g_t$ to shift according to the corresponding velocity component, collectively forming the next MEC embedding $g_{t+1}$. Given that the dimensionality of latent action $z_t$ is smaller than that of $g_t$, $f_{\text{forward}}$ serves as a transformation that combines the abstract latent action with its corresponding MEC embedding to generate a displacement vector $\Delta g_t$. Through this mechanism, the model can transform abstract latent actions into concrete dynamics to predict future states.

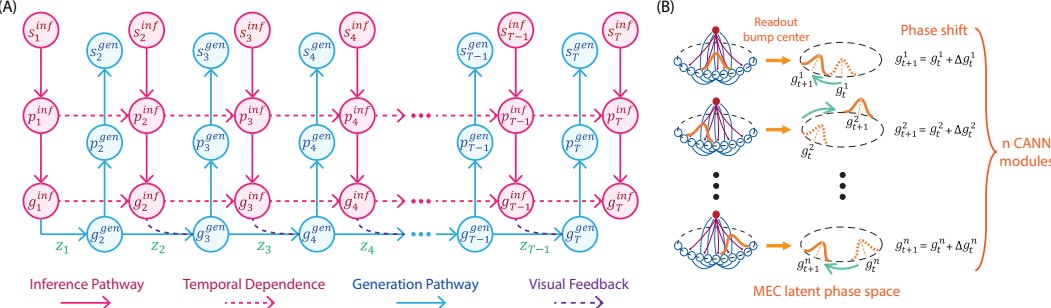

Figure 2: **Overview of the HPC-MEC coupling model.** (A) The graphical model of the HPC-MEC coupling model. The visual inference pathway (solid pink arrow) models the encoding process of $s_{1:T}^{\text{inf}} \to p_{1:T}^{\text{inf}} \to g_{1:T}^{\text{inf}}$. The temporal dependence (dashed pink arrow) ensures the continuity and consistency of the representations. The generation pathway (solid blue arrow) models the transition dynamic and the decoding process of $g_{2:T}^{\text{gen}} \to p_{2:T}^{\text{gen}} \to s_{2:T}^{\text{gen}}$. The visual feedback (dashed purple arrow) can correct the accumulated path integration error. (B) The mechanism of velocity-like abstractions operating in the CANN-inspired MEC latent space.

**The visual feedback.** This model enables autoregressive prediction by providing an initial observation and a sequence of latent actions. Analogous to grid cells in the MEC performing path integration using velocity inputs, our model composes latent actions to predict future states. However, solely relying on path integration inevitably accumulates errors over time. The visual inference pathway addresses this by providing feedback to correct the accumulated errors. Specifically, when visual input is available, the model corrects the current state using the inferred MEC embedding $g_t^{\text{inf}}$ from the observation to predict the next state:

$$g_{t+1}^{\text{gen}} = g_t^{\text{inf}} + f_{\text{forward}}(z_t, g_t^{\text{inf}}). \tag{2}$$

**The separation of contents and structures.** In contrast to earlier world models (Schmidt & Jiang, 2024; Ye et al., 2024; Chen et al., 2024b) that integrate transition dynamics and reconstruction within a single latent space, our HPC-MEC coupling model distinctly encodes contents and structures. This disentanglement promotes feature reuse and efficient representation: the MEC captures shared dynamics across objects, while the HPC retains object-specific information. As a result, the model generalizes transition patterns across visual contexts without conflating appearance with abstract dynamics.

## 3.2 INFERRING LATENT ACTIONS THROUGH AN INVERSE MODEL.

Rather than directly mapping from the high-dimensional observation space, the inverse model infers the latent action $z_t$ from consecutive MEC embeddings $g_t^{\text{inf}}$ and $g_{t+1}^{\text{inf}}$. In the absence of action labels, transitions between embeddings serve as a practical proxy for learning latent actions. We build on this by framing latent actions as transitions on a cognitive map. Our task is thus to learn these abstract latent actions from state transitions to derive an abstract cognitive map. The transitions between MEC embeddings capture the most salient and noise-free dynamics, which the inverse model then distills into a low-dimensional latent action space:

$$z_t = f_{\text{inverse}}(g_{t+1}^{\text{inf}} - g_t^{\text{inf}}), \tag{3}$$

where $f_{\text{inverse}}$ is implemented as a MLP. By operating on the difference between sequential MEC embeddings, $z_t$ captures only the low-dimensional transition dynamics while excluding redundant visual features already encoded in the HPC. The resulting latent actions can be used to predict the next MEC embedding $g_{t+1}^{\text{gen}}$ through path integration, as described in Sec. 3.1.

## 3.3 TRAINING STAGES

We train the inverse model and the HPC-MEC coupling model in a self-supervised learning paradigm. Our input consists solely of video sequences, with the model's objective being to learn latent actions and a world model through self-supervision, thereby eliminating the need for any prior constraints on the abstract space and ground-truth action labels. We employ an alignment loss, adapted from Whittington et al. (2020), to enforce consistency between sensory input and self-motion cues, mimicking the stable spatial coding of the HPC-MEC system. We divide the training process into three stages:

1. **Training the visual inference pathway and the decoding process of the generative pathway:** This stage trains the model to reconstruct observation embeddings to form meaningful HPC embeddings $p_{1:T}^{\text{inf}}$ and MEC embeddings $g_{1:T}^{\text{inf}}$. The training objective combines reconstruction, alignment, and regularization losses:

$$\mathcal{L}_{\text{phase1}} = \mathcal{L}_{\text{reconstruction}}(s_{1:T}^{\text{inf}}, s_{1:T}^{\text{rec}}) + \mathcal{L}_{\text{alignment}}(p_{1:T}^{\text{inf}}, p_{1:T}^{\text{rec}}) + \mathcal{L}_{\text{regularization}}(p_{1:T}^{\text{inf}}, g_{1:T}^{\text{inf}}), \tag{4}$$

   where $s_{1:T}^{\text{rec}}$ and $p_{1:T}^{\text{rec}}$ are the reconstructed observations and HPC embeddings from $g_{1:T}^{\text{inf}}$ respectively. The reconstruction loss and the alignment loss are measured using MSE. The regularization loss encourages the model to learn a structured latent space by minimizing the covariance and variance of the embeddings (Bardes et al., 2022).

2. **Training the inverse model and the transition dynamics of the generative pathway:** Using the meaningful embeddings obtained in stage 1, we train the inverse model to infer latent action $z$ from consecutive MEC embeddings $g^{\text{inf}}$. Simultaneously, we train the transition dynamics to predict the generated MEC embedding $g^{\text{gen}}$. We focus on one-step prediction to avoid accumulated errors from multi-step path integration and prevent the model from learning shortcuts that bypass latent actions. The training objective extends phase 1 with additional alignment and action losses:

$$\mathcal{L}_{\text{phase2}} = \mathcal{L}_{\text{phase1}} + \mathcal{L}_{\text{alignment}}(g_{2:T}^{\text{inf}}, g_{2:T}^{\text{gen}}) + \mathcal{L}_{\text{action}}(f_{\text{forward}}(z_{1:T-1}, g_{1:T-1}^{\text{inf}}), \Delta g_{1:T-1}^{\text{inf}}). \tag{5}$$

3. **Jointly finetuning the HPC-MEC coupling model and the inverse model:** In this final stage, we jointly finetune all model parameters using the phase 2 loss function. Unlike phase 2, the model now learns to autoregressively generate rollouts by performing path integration and forward prediction, using only the $g_1^{\text{inf}}$ and the latent action sequence from the inverse model. This stage can also be regarded as *self-forcing* paradigm (Huang et al., 2025). Detailed descriptions of model training in different stages are discussed in Appendix C.1.

# 4 EXPERIMENTS

**Datasets.** We evaluate our model using three types of datasets. The model is trained on Something-Something v2 (SSv2) dataset (Goyal et al., 2017), a large-scale human activity video dataset including complex interactions with objects without explicit action labels. Several simulated datasets are used to evaluate the out-of-distribution capability of our model, including 3D object rotation/transition datasets (COIL-100 (Nene et al., 1996), MIRO (Kanezaki et al., 2018), *OmniObject3D* (Wu et al., 2023)) and simulated environments (Franka Kitchen (Gupta et al., 2019), Block Pushing (Florence et al., 2022), Push-T (Chi et al., 2023), and LIBERO Goal (Liu et al., 2023).)

## 4.1 ONE-STEP AND AUTOREGRESSIVE PREDICTION

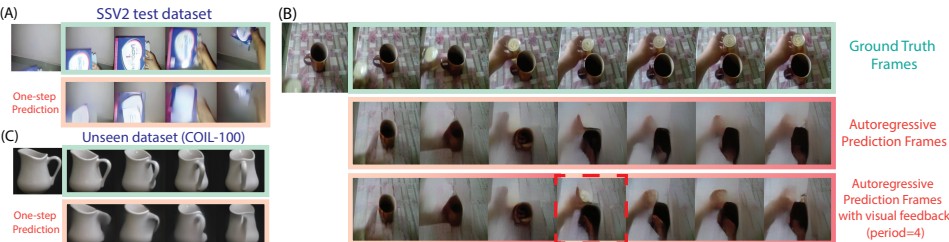

Figure 3: **Evaluation on one-step and autoregressive prediction.** (A) One-step prediction evaluated on the SSv2 test dataset. (B) Autoregressive prediction with and without the visual feedback on the SSv2 test dataset. (C) One-step prediction on an out-of-distribution dataset, COIL-100.

For one-step prediction, the model successfully extracts latent actions from the input video and generates frames that match the dynamics of the ground-truth sequence (Fig. 3(A)). For autoregressive prediction, the model generates the entire sequence by applying a sequence of latent actions to the initial frame. We extend the sequence length to inspect the model's ability in conditional video generation using latent actions, as shown in Fig. 3(B). We find that the model maintains good consistency even when generating longer sequences, although the generation quality gradually decreases over time. This corresponds to the accumulating path integration errors that occur in MEC. Similar to biological systems, the model can correct path integration errors by receiving visual feedback from the sensory input. We test the model's performance after introducing visual feedback at the fourth autoregressive step and observe improved generation quality, with more accurate details in the subsequent frames (Fig. 3(B)). We conduct an additional ablation study to examine the role of latent actions in governing transition dynamics, discussed in Appendix E.

**Generalization to out-of-distribution datasets.** To evaluate whether the model could effectively extract and utilize latent actions to generate OOD scenes, we assess the model on unseen 3D object rotation datasets after pre-training exclusively on the SSv2 dataset. Our results show that the model successfully identifies fundamental rotational transformations as latent actions and generates sequences that closely approximate the ground truth (Fig. 3(C)). The model's ability to generalize to OOD scenes and action distributions is notable, especially considering it was never explicitly trained for the specific task of extracting pure 3D object rotations. We further evaluate the model on simulated benchmarks with significant distributional shifts from human videos, potentially limiting generalization to virtual domains. Full results are provided in Appendix F. Our findings show the model performs robustly in the more naturalistic environments like Franka Kitchen, but less effectively in artificial environments like Push-T.

## 4.2 REUSING ABSTRACT LATENT ACTIONS ACROSS CONTEXTS

We conceptualize abstract latent actions as structural abstractions that can be extracted from video sequences with similar dynamics and reapplied to generate similar movements of different objects or scenes. To validate our model's ability to reuse latent actions, we conduct evaluations using both naturalistic human activity data and simulated datasets.

The results of one-step latent action reuse are shown in Fig. 4(A). We extract the latent action from the purple-highlighted image pairs, which capture hand movements such as disappearing or grabbing

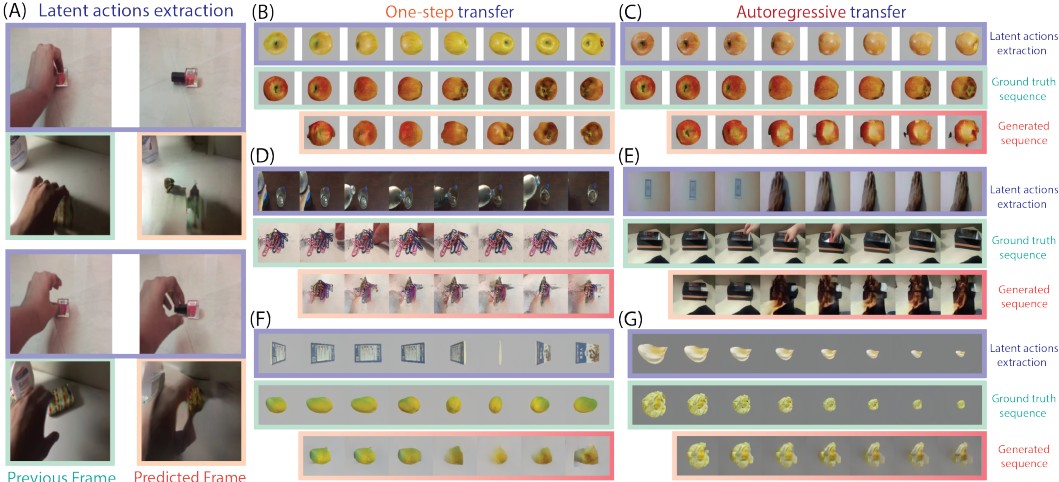

Figure 4: **Latent action transfer.** (A) One-step latent action transfer across different scenes on SSv2. (B)(C) One-step & autoregressive prediction by transferring the sequential latent actions. (D)(E) Autoregressive reuse of latent actions on SSv2. (F)(G) Autoregressive reuse of latent actions on rotation and scaling dynamics across object categories.

objects. We then apply the same latent action to different scenes and generate subsequent frames. The generated frames successfully mimic the dynamics of the original image pairs.

To further investigate the latent action transferability, we conduct experiments applying latent action sequences extracted from one video to frames containing different contexts. Using apple rotation as a case study, we extract latent actions from a sequence featuring a yellow apple and apply them to generate the rotation of a ripe red apple. Our results reveal that while the texture features in the generated sequence correspond to the ripe red apple, the rotational dynamics align with the yellow apple from which the latent actions are derived (Fig. 4(B)). When implementing autoregressive prediction using only the extracted latent actions and the initial frame, we observe that the generated sequence maintains alignment with the rotational dynamics of the source sequence. However, the texture features progressively deviate, becoming more luminous than the ground truth sequence (Fig. 4(C)). We also present two examples to illustrate sequential latent action reuse in human activity scenarios in Fig. 4(D,E). The resulting image sequence captures the dynamics from the source scenario while preserving the content information of the new scene. Fig. 4(F,G) demonstrates two additional cases: rotation and scaling dynamics transfer across object categories. These findings provide compelling evidence that our model successfully decouples and transfers dynamic information while preserving the visual features of the target object.

Table 1: Quantitative comparison of SSIM and LPIPS across models and downstream datasets. Each model performs 8-step sequence generation. * indicates out-of-distribution datasets.

| Dataset | Model | SSIM ↑ | | LPIPS ↓ | |
|---|---|---|---|---|---|
| | | one-step | autoregression | one-step | autoregression |
| SSv2 | LAPA | $0.744_{\pm 0.022}$ | $0.659_{\pm 0.027}$ | $0.357_{\pm 0.017}$ | $0.448_{\pm 0.017}$ |
| | Moto | $0.668_{\pm 0.027}$ | $0.566_{\pm 0.029}$ | $0.301_{\pm 0.022}$ | $0.480_{\pm 0.012}$ |
| | AdaWorld(LAM) | $\mathbf{0.763}_{\pm 0.023}$ | $0.654_{\pm 0.018}$ | $0.295_{\pm 0.026}$ | $0.448_{\pm 0.022}$ |
| | Our model | $0.752_{\pm 0.019}$ | $\mathbf{0.687}_{\pm 0.018}$ | $\mathbf{0.274}_{\pm 0.026}$ | $\mathbf{0.356}_{\pm 0.015}$ |
| COIL-100* | LAPA | $0.589_{\pm 0.038}$ | $0.682_{\pm 0.020}$ | $0.301_{\pm 0.021}$ | $0.385_{\pm 0.016}$ |
| | Moto | $0.700_{\pm 0.031}$ | $0.642_{\pm 0.043}$ | $0.352_{\pm 0.041}$ | $0.473_{\pm 0.031}$ |
| | AdaWorld(LAM) | $0.778_{\pm 0.039}$ | $0.715_{\pm 0.042}$ | $0.312_{\pm 0.048}$ | $0.415_{\pm 0.060}$ |
| | Our model | $\mathbf{0.837}_{\pm 0.021}$ | $\mathbf{0.814}_{\pm 0.032}$ | $\mathbf{0.226}_{\pm 0.034}$ | $\mathbf{0.270}_{\pm 0.031}$ |
| Franka Kitchen* | LAPA | $0.690_{\pm 0.002}$ | $0.532_{\pm 0.003}$ | $0.389_{\pm 0.001}$ | $0.529_{\pm 0.001}$ |
| | Moto(failed) | - | - | - | - |
| | AdaWorld(LAM)(failed) | - | - | - | - |
| | Our model | $\mathbf{0.705}_{\pm 0.004}$ | $\mathbf{0.551}_{\pm 0.005}$ | $\mathbf{0.253}_{\pm 0.003}$ | $\mathbf{0.426}_{\pm 0.003}$ |

**Baselines.** We compare our model with LAPA (Ye et al., 2024), Moto (Chen et al., 2024b), and AdaWorld latent action model(LAM) (Gao et al., 2025), which are the state-of-the-art latent action models. We use two standard metrics: Structural Similarity Index (SSIM) (Wang et al., 2004) for local

structural consistency and Learned Perceptual Image Patch Similarity (LPIPS) (Zhang et al., 2018) for perceptual similarity. As shown in Table 1, our model achieves lower LPIPS scores in both one-step and autoregressive prediction, indicating closer visual similarity with ground-truth dynamics and more effective extraction of latent actions. On the Franka Kitchen dataset, both Moto and AdaWorld LAM fail to produce accurate next-frame predictions, resulting in frames that are almost identical to the preceding frame. Unlike all these baselines, which predict in pixel space, our model operates entirely in latent space and is trained in a self-forcing manner, reducing the computational costs and achieving stronger autoregressive performance. The baselines are also evaluated on the latent action transfer task, but all failed to transfer the abstract structures. All of them produce frames that are identical to the first frame.

**Ablations.** We conduct an ablation study to directly investigate the source of our model's performance. First, we compared our model to a "unified space" alternative in a latent action reuse task. Qualitatively, the unified model shows increasing "texture leakage" from the source video over time, suggesting its latent actions are entangled with content. We quantify this by defining a similarity ratio using features from a pretrained DINOv2 encoder $E$. This ratio $R = \mathbb{E}\left[\frac{cos\_sim(E(\mathbf{o}_t^{\text{gen}}), E(\mathbf{o}_t^{\text{content}}))}{cos\_sim(E(\mathbf{o}_t^{\text{gen}}), E(\mathbf{o}_t^{\text{action}}))}\right]$ measures whether the generated object is closer to the target content or the source content. A higher $R$ indicates stronger alignment with the content sequence and thus better latent action reuse. Second, we replaced our CANN module with a standard state-action concatenation method. This ablated model, along with other VQ-based baselines, all failed the out-of-distribution (OOD) latent action transfer task, unable to apply learned actions to new scenes. The results are shown in Table 2.

Table 2: Quantitative comparison of $R$, SSIM, and LPIPS across ablation models.

| Model | R ↑ | | SSIM ↑ | | LPIPS ↓ | |
|---|---|---|---|---|---|---|
| | one-step | autoregression | one-step | autoregression | one-step | autoregression |
| Our model w/ unified latent space | $2.054_{\pm 0.521}$ | $1.542_{\pm 0.246}$ | $0.901_{\pm 0.007}$ | $0.886_{\pm 0.008}$ | $0.126_{\pm 0.008}$ | $0.179_{\pm 0.008}$ |
| Our model w/o CANN | $2.403_{\pm 0.553}$ | $1.859_{\pm 0.396}$ | $0.894_{\pm 0.022}$ | $0.888_{\pm 0.009}$ | $0.149_{\pm 0.009}$ | $0.177_{\pm 0.010}$ |
| Our model | $\mathbf{3.201_{\pm 0.435}}$ | $\mathbf{2.482_{\pm 0.460}}$ | $\mathbf{0.902_{\pm 0.010}}$ | $\mathbf{0.891_{\pm 0.009}}$ | $\mathbf{0.120_{\pm 0.008}}$ | $\mathbf{0.156_{\pm 0.008}}$ |

## 5 ANALYSIS OF LEARNED LATENT SPACE

Having demonstrated the model's capability to extract and reuse latent actions across various contexts, we now analyze how the hierarchical processing between HPC and MEC embeddings facilitates the emergence of shared structures. We evaluate the SSv2-pretrained model on 3D object rotation datasets, visualizing both HPC and MEC embeddings to gain deeper insights into the model's ability.

**Periodic shared structures.** We first identify objects with periodic rotation patterns and categorize them into three periodicity classes based on texture symmetry: period 1 (360°), $\frac{1}{2}$ (180°), and $\frac{1}{4}$ (90°). Through dimensional reduction analysis using UMAP (McInnes et al., 2018), we observe that these periodicity classes exhibit distinctive low-dimensional trajectories in the embedding space (Fig. 5(A)). The object with period 1 forms a complete circular trajectory, while the object with period $\frac{1}{2}$ forms two overlapping circular trajectories. The object with period $\frac{1}{4}$, with three white sides and one brown side, forms two overlapping small circular trajectories, with the remaining half period forming a larger circular trajectory. Both HPC and MEC embeddings exhibit similar periodicity patterns, but MEC representations form more clearly defined shared rotation features. To verify this, we perform multi-class analyses to distinguish between two embedding spaces.

**In-class shared structures.** To analyze in-class shared structures, we examine three object categories: pumpkins, red apples, and yellow apples, each containing multiple instances. We process rotation sequences of each object using our model to extract HPC and MEC embeddings, and visualize these representations with UMAP. The results (Fig. 5(B)) demonstrate that while both HPC and MEC exhibit inter-category separation, $\boldsymbol{p}^{\text{inf}}$ additionally shows clear intra-category differentiation. Specifically, for different in-class objects, $\boldsymbol{p}^{\text{inf}}$ forms separated circular trajectories, reflecting object-wise features. In contrast, $\boldsymbol{g}^{\text{inf}}$ and $\boldsymbol{g}^{\text{gen}}$ trajectories substantially overlap, capturing the in-class shared rotational features. These shared structures propagate through the generative pathway, shape the manifold of $\boldsymbol{p}^{\text{gen}}$, making rotational features more salient in the high-dimensional space.

To quantify the in-class structural sharing, we train separate lightweight decoders to classify object categories from HPC and MEC embeddings. Test objects are excluded during training, so higher test accuracy indicates greater structural consistency within object classes. Our results show that after

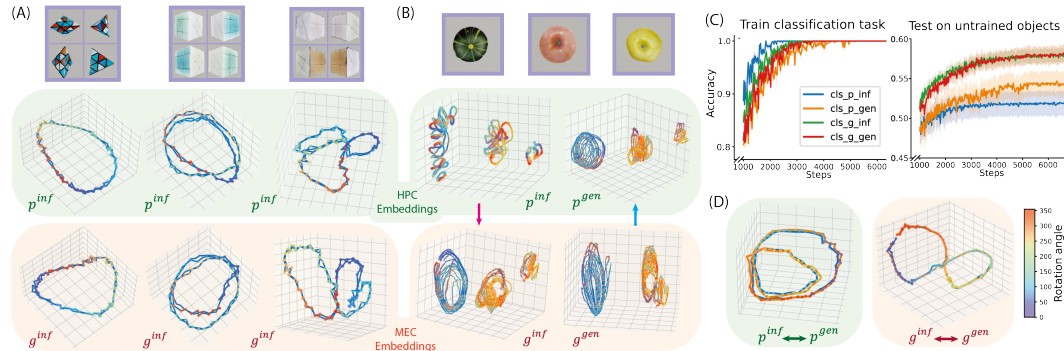

Figure 5: **Analysis of HPC and MEC embeddings.** (A) UMAP visualization of HPC and MEC embeddings grouped by periodicity class. Each object completes two full rotations. (B) UMAP visualization of HPC and MEC embeddings grouped by object category. (C) Classification accuracy of object categories using HPC and MEC embeddings. (D) Alignment between inference and generation embeddings for an individual object.

training convergence, the test accuracy on $p^{\text{inf}}$ is consistently lower than $g^{\text{inf}}$ and $g^{\text{gen}}$, indicating that HPC embeddings encode fewer in-class shared structures. In contrast, MEC embeddings better capture category patterns, enhancing generalization to novel instances (Fig. 5(C)). Despite discrepancies between the UMAP reductions of $p^{\text{inf}}$ and $p^{\text{gen}}$, visualizing individual objects' inference and generation embeddings shows high consistency in both HPC and MEC, validating the model's training objectives and the alignment between inference and generative pathways (Fig. 5(D)).

**Generalization to robotic dynamics.** We analyze sequences of the same action (e.g., "opening the upper cabinet" in Franka Kitchen) performed under varying contexts (e.g., different object positions or microwave states). We then compute the cosine similarity of the state transitions within the HPC space ($\Delta p$), the MEC space ($\Delta g$), and the latent action space ($z$). The results are shown in Table 3. We can see that latent action trajectories share a more similar pattern than other state transitions. These results suggest that the model can extract content-independent structures from simulated environments.

Table 3: Sequence similarity of different latent space transitions

| Transitions | $z$ | $\Delta g^{\text{inf}}$ | $\Delta g^{\text{gen}}$ | $\Delta p^{\text{inf}}$ | $\Delta p^{\text{gen}}$ |
|---|---|---|---|---|---|
| Cos Sim ↑ | $\mathbf{0.235 \pm 0.021}$ | $0.146 \pm 0.063$ | $0.152 \pm 0.057$ | $0.024 \pm 0.061$ | $0.114 \pm 0.056$ |

## 6 DISCUSSION AND LIMITATIONS

Our finding shows how abstract latent actions can be effectively extracted from real-world video sequences while maintaining meaningful disentangled hierarchical latent spaces. The combination of the inverse model with the HPC-MEC-inspired world model enables efficient disentangling of abstract structures from specific contents, facilitating robust transfer capabilities. Our analysis of the HPC and MEC representations further highlights the potential for neuro-inspired models to encode and reuse abstract structures from real-world transition dynamics. The capability of our model to reuse these latent actions across diverse contexts underscores the critical role of structural generalization in embodied agents. Our model exhibits generalization capabilities across different action distributions, thereby paving the way for leveraging latent action learning to achieve generalizable embodied agents. We also streamlined the dynamics of the CANN-based path integration, facilitating the future scaling of the model.

**Limitations.** Our model has several limitations. First, it is susceptible to autoregressive compounding errors, which could potentially be mitigated by incorporating a memory bank. Second, performance degrades on benchmarks that exhibit substantial distributional drift from the training data (human videos); increasing the diversity of the training dataset may alleviate this issue. Additionally, coordinating multiple independent entities remains a major challenge. Future work will explore hierarchical HPC-MEC structures or object-centric representations to tackle this problem.

## ETHICS STATEMENT

While our foundational research can benefit embodied AI, the ability to manipulate and reuse motion dynamics carries risks. These include the potential for malicious fake content generation and the possibility that the model could amplify societal biases from its training data, leading to unsafe or unfair behavior. To mitigate these risks, we recommend implementing content authentication mechanisms, establishing access controls for model deployment, and conducting regular bias audits on training data and outputs. We advocate for transparent reporting of model limitations and responsible use guidelines to ensure ethical applications in downstream tasks.

## REPRODUCIBILITY STATEMENT

We are committed to ensuring the reproducibility of our results and have taken several measures to facilitate replication. In the supplementary materials, we provide comprehensive code for model training to enable others to reproduce our results. Upon completion of the review process, we will also release pre-trained models. In Section 3, we introduce the model architecture and training stages. In Appendix B, we provide detailed design motivations, comprehensive model specifications, parameters, and sensitivity analysis. In Appendix C, we detail loss functions, hyperparameters, and computational requirements. In Appendix D, we provide complete dataset descriptions. In Appendix G, we detail the experimental setup for latent action space analysis.

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

## A LARGE LANGUAGE MODELS STATEMENT

In this work, the research ideas and literature review are conducted independently without the use of large language models. For writing, we use LLMs to polish the language and condense certain paragraphs, followed by careful human review and revision.

## B MODEL DETAILS

### B.1 MODEL DESIGN MOTIVATION

Our model is a brain-inspired framework guided by neuroscience and implemented with state-of-the-art deep learning modules.

Separating a content pathway (HPC) from a structure pathway (MEC) is directly inspired by established theories in computational neuroscience. Using a latent action space learned via an inverse dynamics model is a common and effective framework adopted by many baselines in this field for learning from unlabeled video. Our work builds upon this solid foundation. Within this framework, our component choices are deliberate. The MEC's path integration role is implemented using a Continuous Attractor Neural Network (CANN), a classic computational model of grid cells (Gardner et al., 2022; McNaughton et al., 2006; Fuhs & Touretzky, 2006; Burgess et al., 2007). Latent actions are inferred via an inverse dynamics model, a standard approach rooted in theories of the cerebellum (Wolpert et al., 1998).

Finally, we choose specific state-of-the-art implementations for these functional roles to ensure high performance. The pretrained visual multi-scale VQ-VAE (Tian et al., 2024) provides a stable and high-quality visual representation, analogous to processed input from the visual cortex. The Spatial-Temporal Transformer (Bruce et al., 2024; Ye et al., 2024) models complex dependencies across space and time, exactly what is required to integrate the content and structure signals to predict future states.

### B.2 MODEL PARAMETER

We provide the model parameters (Table 4) used in our experiments.

### B.3 PRETRAINED ENCODER DETAILS

We use the pretrained multi-scale VQ-VAE (depth=16) from the VAR model to extract visual features. The feature map we use, $\hat{f}$, is the sum of outputs from the multi-scale vector quantization layers. This feature map is fed directly into our model. This design is intentional: it simulates how the HPC-MEC circuit receives pre-processed information from the visual cortex, and it frames the task as a prediction problem entirely within the latent space (akin to JEPA (LeCun, 2022)), meaning our model does not perform pixel-level reconstruction.

### B.4 CANN DYNAMICS AND PATH INTEGRATION

Inspired by the role of grid cells in the MEC, we model transition dynamics using a continuous attractor neural network (CANN). CANN is a specific type of recurrent neural network designed to maintain a stable, continuous representation of information in its activity pattern. Unlike classical attractor networks that store discrete, unstructured patterns, CANNs are distinguished by their ability to encode structured patterns organized by metric relationships. This geometric structure allows a CANN to maintain a system's state as a stable, localized "activity bump". Path integration is achieved by using the inferred latent action as a velocity operator that controllably shifts this bump along the network's metric axes. This ability to maintain a stable state and systematically transform it via a velocity input is what allows the CANN to integrate a path and predict the next state's structure.

In our model, the MEC embeddings $\boldsymbol{g}$ represent abstract structures in high-dimensional space (Klukas et al., 2020). When multiple one-dimensional CANNs are combined, their joint state space forms an $N$-dimensional continuous manifold (Burgess & Burgess, 2014). This modular representation enables the decomposition of spatial dynamics, where each module is driven by latent action inputs.

Table 4: Model parameters

| COMPONENT/PARAMETER | VALUE |
| --- | --- |
| **Input parameters** | |
| Input Channels | 3 |
| Input Image Height | 256 |
| Input Image Width | 256 |
| VQ-VAE Encoder Depth | 16 |
| VQ-VAE Encoder Feature Map Channels | 32 |
| VQ-VAE Encoder Feature Map Heights | 16 |
| VQ-VAE Encoder Feature Map Widths | 16 |
| Patch Size | 4 |
| Patch Height | 4 |
| Patch Width | 4 |
| | |
| **HPC Model** | |
| HPC Hidden Size (total) | 8192 |
| Per-patch Hidden Dimension | 512 |
| Spatial Transformer Depth | 4 |
| Temporal Transformer Depth | 4 |
| | |
| **MEC Model** | |
| MEC Hidden Size (total) | 4096 |
| Per-patch Hidden Dimension | 256 |
| Spatial Transformer Depth | 4 |
| Temporal Transformer Depth | 4 |
| | |
| **Inverse World Model** | |
| Action Dimension | 2048 |
| Per-patch Action Dimension | 128 |
| $f_{\text{inverse}}$ Residual Blocks | 2 |
| $f_{\text{forward}}$ Residual Blocks | 4 |

We formalize the CANN dynamics implemented in our MEC module for encoding abstract representations and performing path integration. Specifically, we draw connections between the classical differential equation-based formulation of CANNs and our discrete-time, learnable implementation.

The canonical dynamics of a one-dimensional CANN are described by the following first-order differential equation:

$$\tau \frac{d\mathbf{u}(t)}{dt} = -\mathbf{u}(t) + \mathbf{W}_r \mathbf{r}(t) + \mathbf{W}_{\text{in}} \mathbf{v}(t), \tag{6}$$

where:

- $\mathbf{u}(t)$ is the hidden state of the network;

- $\mathbf{r}(t)$ is the instantaneous neural firing rate, which is derived from the hidden state using a non-linear activation function $\mathbf{r}(t) = H(\mathbf{u}(t))$;

- $\mathbf{v}(t)$ is the external motion input (e.g., velocity);

- $\mathbf{W}_r$ and $\mathbf{W}_{\text{in}}$ are the recurrent and input weight matrices, respectively;

- $\tau$ is a time constant.

This formulation supports both self-sustaining activity through recurrent feedback and perturbation by external motion cues. To enable numerical modeling, Equation equation 6 is discretized using forward Euler integration:

$$\mathbf{u}_{t+1} = (1 - \alpha)\mathbf{u}_t + \alpha \left( \mathbf{W}_r \mathbf{r}_t + \mathbf{W}_{\text{in}} \mathbf{v}_t \right), \tag{7}$$

where $\alpha = \Delta t / \tau$.

For simplicity, we omit the explicit modeling of Gaussian bump profiles and instead use a nonlinear function to extract bump centers: $\mathbf{g}(t) = \sigma(\mathbf{r}(t))$. Rather than explicitly designing $\mathbf{W}_r$, we learn the continuous attractor dynamics through a temporal attention mechanism and impose structural regularity using a second-order temporal smoothing loss on $\mathbf{g}$. This loss penalizes high acceleration

in the latent space and is given by:

$$\mathcal{L}_{\text{smooth}} = \frac{1}{B(T-2)} \sum_{b=1}^{B} \sum_{t=1}^{T-2} \left\| \mathbf{g}_{b,t+2} - 2\mathbf{g}_{b,t+1} + \mathbf{g}_{b,t} \right\|^2 , \tag{8}$$

where $B$ is the batch size and $T$ is the sequence length. The temporal attention with causal masking integrates the historical information into the current state, encouraging low curvature in the grid trajectory, reflecting the physical intuition that continuous movement through space should induce smooth changes in internal state.

Focusing on the evolution of bump centers rather than firing rates, we implement the path integration update as:

$$\mathbf{g}_{t+1} = \mathbf{g}_t + f(\mathbf{g}_t, \mathbf{z}_t), \tag{9}$$

where $\mathbf{z}_t$ denotes the latent action and $f(\cdot)$ is a learnable function that models the dynamics induced by $\mathbf{z}_t$. The recurrent dynamics of $\mathbf{g}$ are embedded within the temporal attention structure to maintain continuity over time and to preserve the attractor manifold structure.

### B.5 SENSITIVITY ANALYSIS

For the number of CANN modules (MEC dimension), our current model uses 4096 modules (256 per visual patch). Reducing this to 2048 still allows the model to encode scene-specific dynamics, but with a noticeable loss of detail and increased blurriness in the generated images. A further reduction to 1024 exacerbates this issue and leads to convergence difficulties during training.

Regarding the latent action dimension, we currently use a dimension of 2048. We find that the model can still predict the next frame with a dimension of 1024, though the generation quality is compromised. Compressing the latent action dimension further makes convergence very difficult, often causing the model to learn a trivial solution where it simply outputs the previous frame as its prediction.

### B.6 VISUAL FEEDBACK DETAILS

Since path integration alone accumulates errors over time, the visual inference pathway provides corrective feedback to mitigate this drift. Specifically, in the absence of visual feedback, the model predicts the next state via path integration based on the current prediction $\boldsymbol{g}_t^{gen}$:

$$\boldsymbol{g}_{t+1}^{gen} = \boldsymbol{g}_t^{gen} + f_{\text{forward}}(\boldsymbol{z}_t, \boldsymbol{g}_t^{gen}) \tag{10}$$

When visual input is available, the model corrects the current state using the inferred MEC embedding $\boldsymbol{g}_t^{inf}$ from the observation to predict the next state:

$$\boldsymbol{g}_{t+1}^{gen} = \boldsymbol{g}_t^{inf} + f_{\text{forward}}(\boldsymbol{z}_t, \boldsymbol{g}_t^{inf}) \tag{11}$$

This update does not introduce instability because $\boldsymbol{g}_t^{inf}$ and $\boldsymbol{g}_t^{gen}$ lie in the same latent space. The alignment loss $\mathcal{L}_{\text{alignment}}(\boldsymbol{g}_{2:T}^{\text{inf}}, \boldsymbol{g}_{2:T}^{\text{gen}})$ not only trains the inverse model but also ensures that replacing $\boldsymbol{g}_t^{gen}$ with $\boldsymbol{g}_t^{inf}$ during feedback remains stable. Moreover, visual feedback is used only after the model has been sufficiently trained, at which point the two embeddings are well aligned. As a result, even though the model is not trained on long sequences, the correction mechanism allows it to autoregressively generate accurate predictions over time.

## C  MODEL TRAINING DETAILS

### C.1  LOSS FUNCTIONS

We elaborate on the loss functions designed for each training phase in detail:

- **The reconstruction loss:** We include losses between observation embeddings and three different generative model predictions to accelerate learning. The reconstructed observation embeddings $s^{\text{recon}}$ can be generated directly from the inferred $p^{\text{inf}}$ or $g^{\text{inf}}$ embeddings, while $s^{\text{gen}}$ is generated from the generative pathway. In phase 1, we use $p_{1:T}^{\text{inf}} \to s_{1:T}^{\text{recon}}$ and $g_{1:T}^{\text{inf}} \to p_{1:T}^{\text{inf}} \to s_{1:T}^{\text{recon}}$ to compute the reconstruction loss. In phases 2 and 3, the transition dynamics of the model predict the next generated observation embeddings $g_{2:T}^{\text{gen}} \to p_{2:T}^{\text{gen}} \to s_{2:T}^{\text{gen}}$, and we use $s_{2:T}^{\text{gen}}$ to compute the reconstruction loss.

- **The alignment loss:** The $\mathcal{L}_{\text{alignment}}$ is used to align the latent representations from inference and generation. This loss is inspired by TEM (Whittington et al., 2020), where the inferred HPC embeddings $p^{\text{inf}}$ are aligned with the generated HPC embeddings $p^{\text{gen}}$, and the inferred MEC embeddings $g^{\text{inf}}$ are aligned with the generated MEC embeddings $g^{\text{gen}}$.

- **The action loss:** The $\mathcal{L}_{\text{action}}$ constrains the predicted displacement vector $\Delta g_t^{\text{gen}} = f_{\text{forward}}(z_t, g_t^{\text{gen}})$ to be close to the true displacement vector $\Delta g_t^{\text{inf}}$ at time step $t$. The action loss is computed as the mean squared error (MSE) between the predicted and true displacement vectors. Cosine similarity is also used to ensure the predicted displacement vector aligns with the true displacement vector. To prevent the model from falling into a local minimum during training, where the $g$ alignment loss might make the predicted $g_{t+1}^{\text{gen}}$ closer to $g_t^{\text{inf}}$ rather than $g_{t+1}^{\text{inf}}$, we add an extra contrastive loss term that constrains the distance between predicted $g_{t+1}^{\text{gen}}$ and $g_t^{\text{inf}}$. This is implemented using cosine similarity. The overall form of the loss is:

$$\mathcal{L}_{\text{action}} = \text{MSE}(\Delta g_{1:T-1}^{\text{gen}}, \Delta g_{1:T-1}^{\text{inf}}) + \alpha \cdot \left(1 - \text{CosSim}(\Delta g_{1:T-1}^{\text{gen}}, \Delta g_{1:T-1}^{\text{inf}})\right) + \beta \cdot \text{CosSim}(g_{1:T-1}^{\text{gen}}, g_{1:T-1}^{\text{inf}}), \tag{12}$$

where $\alpha$ and $\beta$ are hyperparameters that control the relative importance of the cosine similarity terms.

- **The regularization loss:** To prevent collapse, we utilize the VICReg objectives (Bardes et al., 2022) to regularize $p_t^{\text{inf}}$ and $g_t^{\text{inf}}$. The variance loss encourages the model to maintain a certain level of variance across batches in the latent space, while the covariance loss penalizes the model for having high covariance between different dimensions of the latent space. The overall form of the regularization loss is:

$$\mathcal{L}_{\text{var}}(Z, \gamma) = \frac{1}{TD} \sum_{t=0}^{T} \sum_{j=0}^{D} \max\left(0, \gamma - \sqrt{\text{Var}(Z_{:,t,j}) + \varepsilon}\right) \tag{13}$$

$$\mathcal{L}_{\text{variance}} = \mathcal{L}_{\text{var}}(g^{\text{inf}}, \gamma = 0.5) + \mathcal{L}_{\text{var}}(p^{\text{inf}}, \gamma = 0.5) \tag{14}$$

$$\mathcal{L}_{\text{cov}}(Z) = \frac{1}{D(D-1)} \sum_{i \neq j} (\text{Cov}(Z)_{ij})^2 \tag{15}$$

$$\mathcal{L}_{\text{covariance}} = \mathcal{L}_{\text{cov}}(g^{\text{inf}}) + \mathcal{L}_{\text{cov}}(p^{\text{inf}}) \tag{16}$$

$$\mathcal{L}_{\text{regularization}} = \phi \cdot \mathcal{L}_{\text{variance}} + \psi \cdot \mathcal{L}_{\text{covariance}} \tag{17}$$

where $\phi$ and $\psi$ are hyperparameters that control the relative importance of the variance and covariance losses, respectively.

### C.2  TRAINING HYPERPARAMETERS

We provide the training hyperparameters (Table 5) used in our experiments.

### C.3  COMPUTE REQUIREMENTS

The model is trained on a large dataset (SSV2, 220,000 videos), with training requiring 6-8 hours (10 epochs, parallel training using 3 A100 GPUs). Inference time is very fast due to the relatively small

Table 5: Training hyperparameters

| PHASE | PARAMETER | VALUE |
|---|---|---|
| **Fixed parameters** | | |
| | Learning Rate | 1e-4 |
| | Optimizer | AdamW |
| | Weight Decay | 1e-4 |
| | Betas | (0.9, 0.999) |
| | Gradient Clipping | 0.1 |
| | lr_scheduler | CosineAnnealingLR |
| | Epochs | 10 |
| **Stage 1** | | |
| | Batch Size | 32 |
| | Sequence Length | 8 |
| | $\mathcal{L}_{\text{recon}}^{\boldsymbol{p}^{\text{inf}} \to \boldsymbol{s}^{\text{rec}}}$ Weight | 5.0 |
| | $\mathcal{L}_{\text{recon}}^{\boldsymbol{g}^{\text{inf}} \to \boldsymbol{s}^{\text{rec}}}$ Weight | 5.0 |
| | $\mathcal{L}_{\text{alignment}}^{\boldsymbol{p}}$ Weight | 0.22 |
| | $\phi$ (variance loss) | 0.01 |
| | $\psi$ (covariance loss) | 0.01 |
| **Stage 2/3** | | |
| | Batch Size | 168/32 |
| | Sequence Length | 2/8 |
| | $\mathcal{L}_{\text{recon}}^{\boldsymbol{p}^{\text{inf}} \to \boldsymbol{s}^{\text{rec}}}$ Weight | 5.0 |
| | $\mathcal{L}_{\text{recon}}^{\boldsymbol{g}^{\text{inf}} \to \boldsymbol{s}^{\text{rec}}}$ Weight | 5.0 |
| | $\mathcal{L}_{\text{gen}}^{\boldsymbol{g}^{\text{gen}} \to \boldsymbol{s}^{\text{gen}}}$ Weight | 3.0 |
| | $\mathcal{L}_{\text{alignment}}^{\boldsymbol{p}}$ Weight | 1.0 |
| | $\mathcal{L}_{\text{alignment}}^{\boldsymbol{g}}$ Weight | 5.0 |
| | $\alpha$ (action loss) | 1 |
| | $\beta$ (contrastive loss) | 1 |
| | $\phi$ (variance loss) | 0.05 |
| | $\psi$ (covariance loss) | 0.05 |

size of the spatial-temporal transformer and multi-scale VQ-VAE, resulting in minimal overhead. So far, increasing model size has not led to significant time increases.

# D  DATASET DETAILS

We aim to learn abstract latent actions from real-world videos. Unlike 2D games or robot demonstrations, real-world human videos exhibit diverse transitions without explicit action labels. We investigate whether pre-training on large-scale human video datasets enables our model to learn versatile latent actions that generalize to unseen data. We use the following datasets.

## D.1  SOMETHING-SOMETHING V2

Something-Something V2(SSV2) (Goyal et al., 2017) contains 220,847 video clips of humans performing actions with everyday objects. We use these large-scale real-world human videos to train our model and maintain the same train/validation/test splits as established in Goyal et al. (2017).

## D.2  3D OBJECTS ROTATION DATASETS

We use three different rotation datasets to evaluate and analyse the model. COIL-100 (Nene et al., 1996) contains images of 100 objects viewed from different angles. MIRO (Kanezaki et al., 2018) is another dataset of 3D object rotations along a different axis.

We also create a synthetic dataset of 3D object rotations, referred to as *OmniRotation*, containing 5911 objects of 216 daily categories with 72 different views per object. We use Blender (Community, 2018) to render meshes from the OmniObject3D (Wu et al., 2023), a dataset of high-quality real-scanned meshes, to create 3D rotation objects. Each object mesh is initialized at $0°$ and then rotated $360°$ around the vertical axis in $5°$ increments, yielding 72 rendered views per object. Our dataset covers 216 categories with a long-tailed distribution, incorporating most daily object realms. We include all raw scans provided on the official website, where the number of categories and objects may slightly differ from those reported in the original OmniObject3D paper. The rendering code is adapted from the implementation provided by Deitke et al. (2023).

## D.3  SIMULATED BENCHMARKS

We also evaluate our model on simulated benchmarks to investigate whether the model trained on real-world data can transfer to virtual environments. We use four different simulated datasets: Franka Kitchen (Gupta et al., 2019), Block Pushing (Florence et al., 2022), Push-T (Chi et al., 2023), and LIBERO Goal (Liu et al., 2023).

## D.4  SEQUENCE CONSTRUCTION FROM THE 3D OBJECTS ROTATION DATASETS

Since the model takes image sequences as input rather than single object views, we need to construct sequences using images from the 3D object rotation datasets. We use three 3D object rotation datasets. Here, we describe how to construct sequences from different object views in these datasets.

Each sequence consists of frames of a single object. Between any two adjacent frames, the second frame is a rotated version of the first. The action here corresponds to the rotation angle: a positive value indicates clockwise rotation, while a negative value indicates counterclockwise rotation. Therefore, we construct a sequence by defining an initial frame and a sequence of rotation actions; the corresponding images are retrieved from the rotation datasets.

Here are some experiments' sequence construction settings:

- In Section 4.1 Fig. 3(C), the relative rotation actions are fixed at $5°$, objects in COIL-100 are rotated clockwise around the vertical axis by $5°$ per frame.
- In Section 4.2, we use different fixed parameters: Fig. 4(B), (C), and (F) use rotation angles of $30°$, $20°$, and $30°$ per step respectively; Fig. 4(G) uses fixed scaling of $0.85$ per step.
- In Section 4.2 Table 1, the relative rotation actions are randomly sampled from $-90°$ to $90°$.
- In Section 4.2 Table 2, the relative rotation actions are randomly sampled from $-30°$ to $30°$.

# E ABLATION STUDY ADDITIONAL RESULTS

## E.1 LATENT ACTION VALIDITY EXPERIMENT

We conduct another ablation study to examine the role of latent actions in governing transition dynamics. Recall that the transition dynamics in our model are defined as:

$$\mathbf{g}_{t+1} = \mathbf{g}_t + f_{\text{forward}}(\mathbf{z}_t, \mathbf{g}_t), \tag{18}$$

where $\mathbf{z}_t$ denotes the latent action, and $f_{\text{forward}}$ integrates content information into $\mathbf{z}_t$ to generate the displacement vector $\Delta\mathbf{g}_t$. Our ablation study focuses on two aspects: the latent action and content binding. First, we disrupt the input to the inverse model by setting it to zero, and then perform both one-step and autoregressive predictions (Fig. 6(A, B)). We observe that in the one-step prediction, the model collapses to simply copying the previous frame, while in the autoregressive setting, only the first frame is retained, and the sequence shows no meaningful transitions. Second, we impair the content binding by allowing $\mathbf{z}_t$ to be combined only with a zero input to produce $\Delta\mathbf{g}_t$ (Fig. 6(C, D)). In this case, one-step prediction generally preserves the overall transition dynamics, but the generated details are degraded; by comparison, autoregressive prediction yields even poorer results. These findings indicate that the latent action primarily drives the main transitions, whereas content binding is essential for reconstructing detailed, scene-specific information.

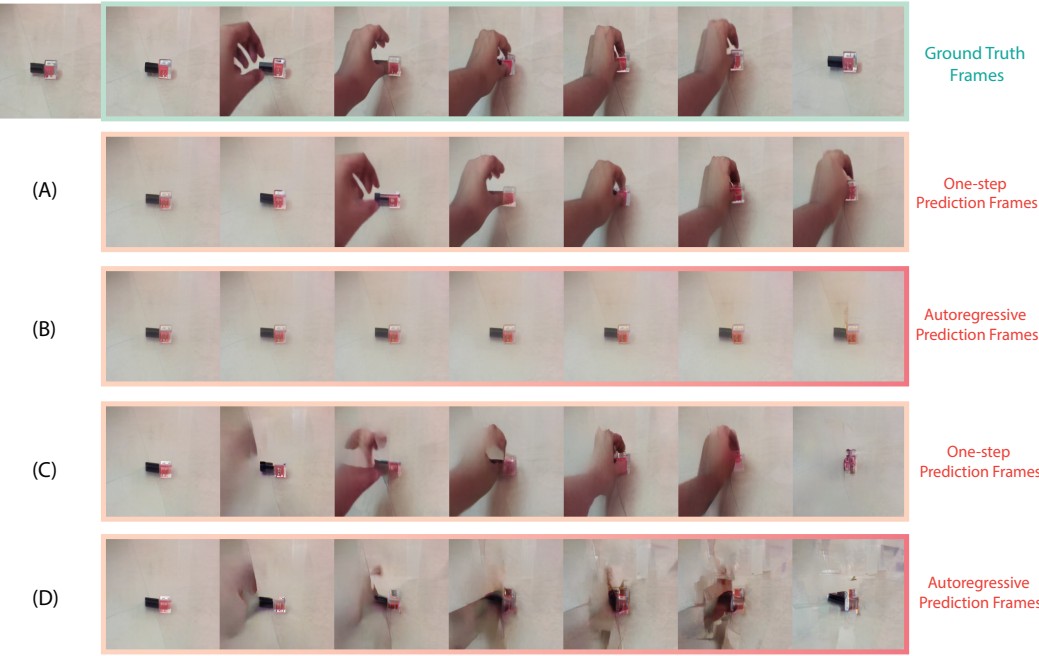

Figure 6: **Latent action validity experiment.** (A) The inverse model receives zero inputs, resulting in meaningless latent actions for one-step prediction. (B) Meaningless latent action for autoregression. (C) $f_{\text{forward}}$ combines latent actions with meaningless content information and performs one-step prediction. (D) Meaningless content information binds to latent actions and performs autoregression.

# F    ADDITIONAL RESULTS

## F.1    PREDICTION RESULTS

### F.1.1    ONE-STEP PREDICTION IN OUT-OF-DISTRIBUTION ROTATION DATASETS

We provide more visualizations of the model's prediction on several rotation datasets. The results are shown in Fig. 7.

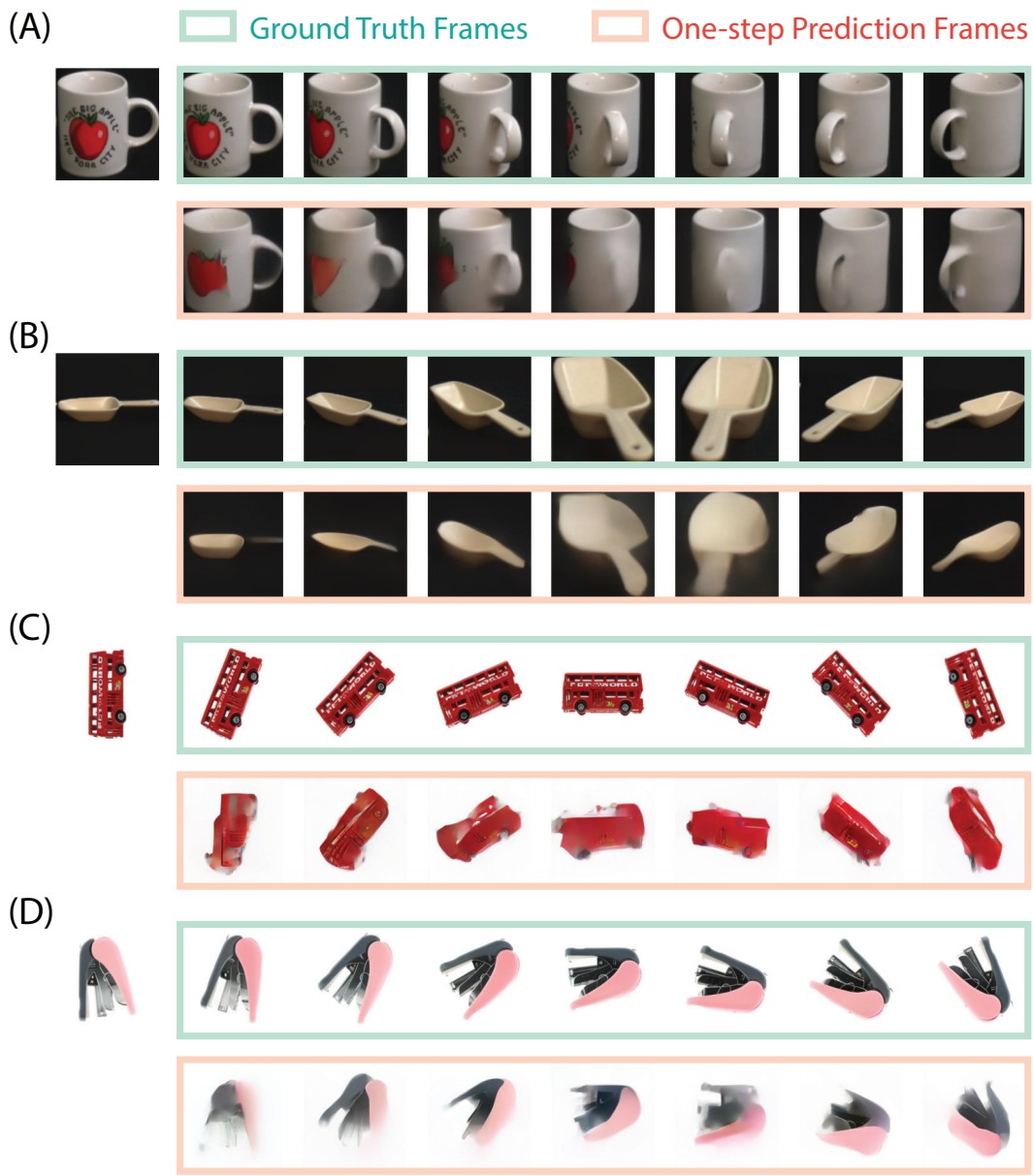

Figure 7: **One-step prediction in rotation datasets.** (A, B) One-step prediction evaluated on the COIL-100 dataset. (C, D) One-step prediction evaluated on the MIRO dataset.

### F.1.2 One-step prediction in out-of-distribution simulated environments

We provide one-step prediction results on simulated benchmarks with substantial distributional shifts from human videos in Fig. 8. The model performs robustly in the more naturalistic Franka Kitchen (Gupta et al., 2019), but less effectively in artificial environments like Push-T (Chi et al., 2023) and Block Pushing (Florence et al., 2022).

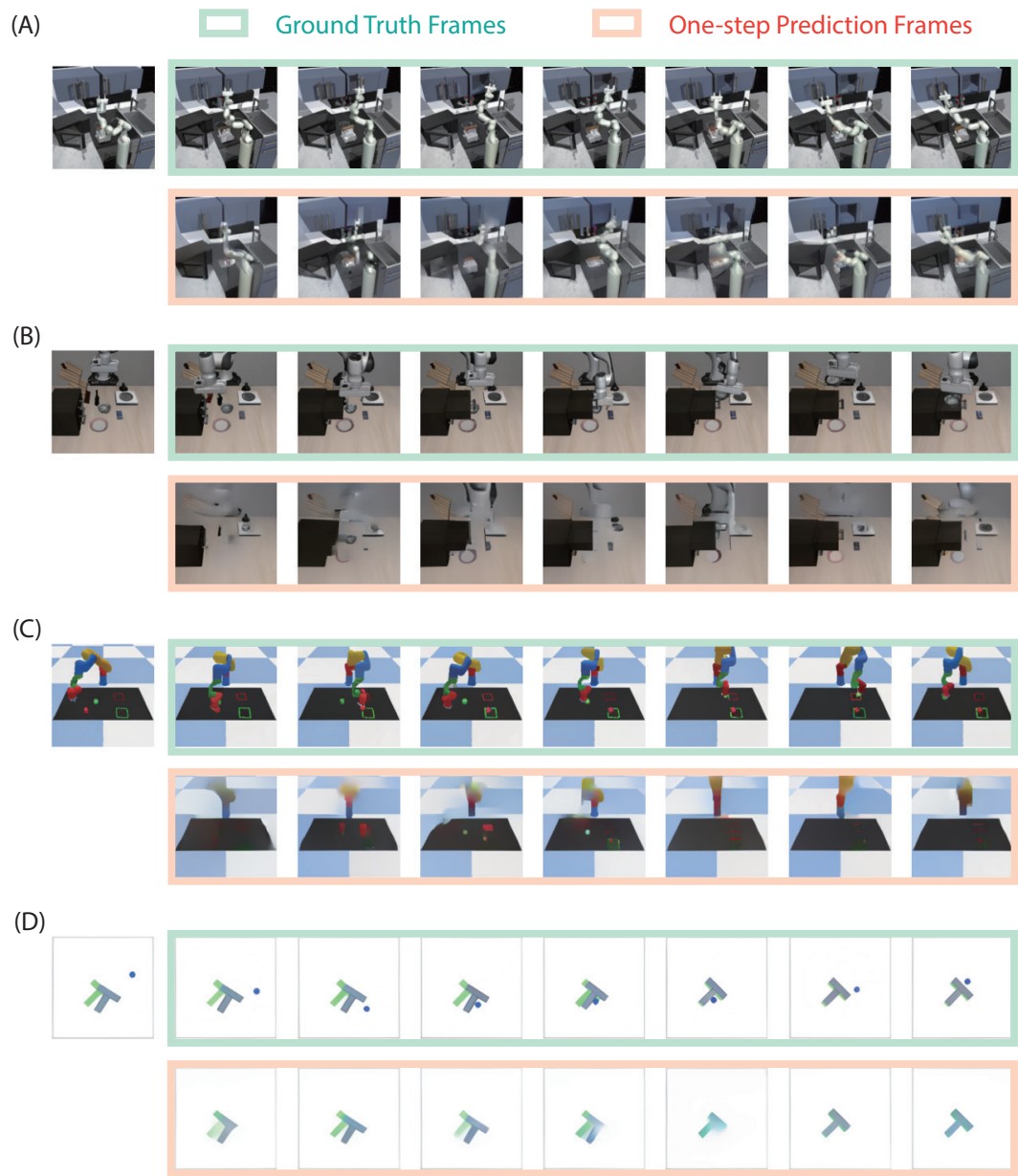

Figure 8: **One-step prediction in simulated environments.** (A) One-step prediction evaluated in Franka Kitchen. (B) LIBERO Goal. (C) Block Pushing. (D) Push-T.

### F.2 LATENT ACTION REUSE RESULTS

#### F.2.1 ONE-STEP AND AUTOREGRESSIVE REUSE OF LATENT ACTIONS ON UNSEEN OMNIROTATION

We present additional results on OmniRotation to validate robustness in Fig. 9. Here we examine cubic objects, where latent actions from a source sequence effectively transform frames of a different object, aligning well with the source's rotational dynamics.

(A)

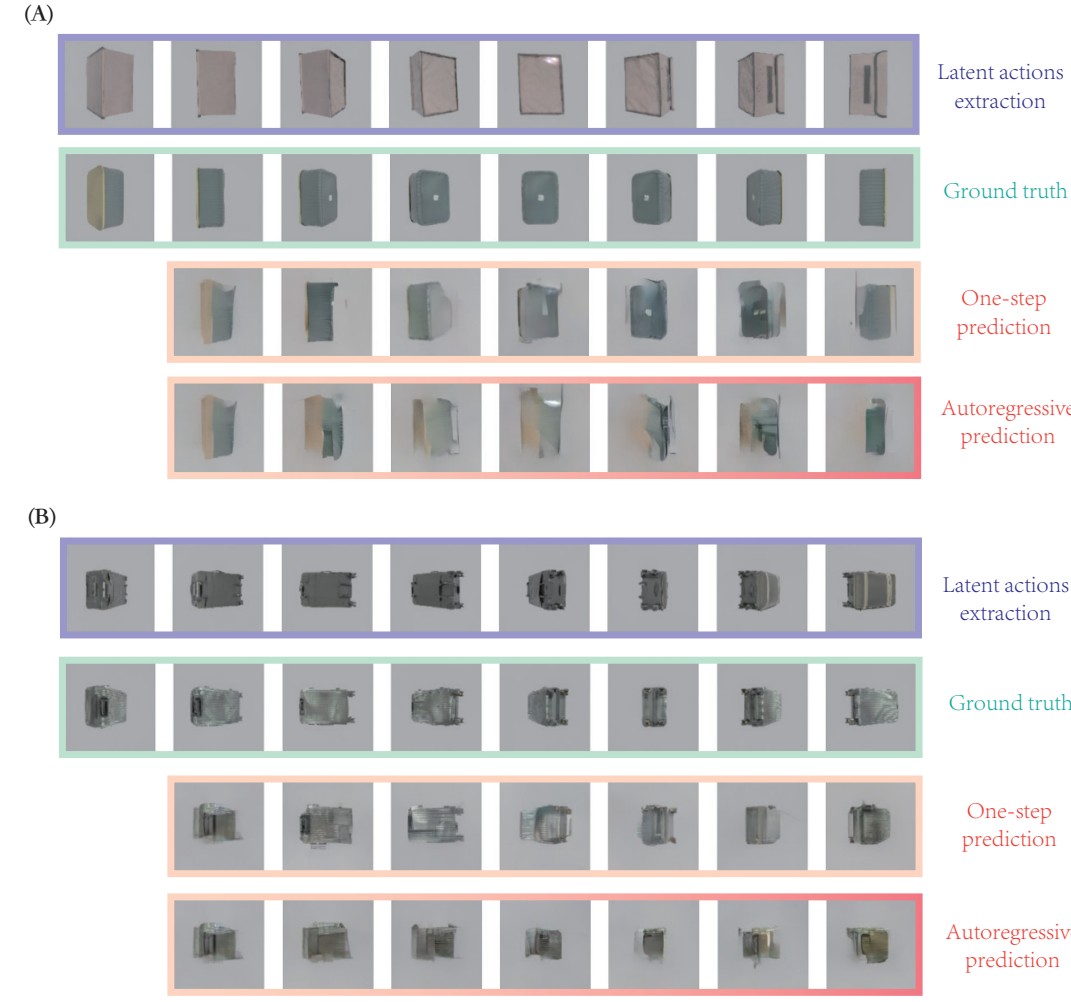

(B)

Figure 9: **Latent action transfer on OmniRotation.** (A, B) Two examples demonstrating one-step and autoregressive reuse of latent actions for cubic objects in the OmniRotation dataset.

#### F.2.2 ONE-STEP AND AUTOREGRESSIVE REUSE OF LATENT ACTIONS ON UNSEEN ARTIFICIAL ENVIRONMENT FRANKA KITCHEN

In Section 5, we demonstrate the model's ability to extract shared latent actions from sequences of the same action performed under varying contexts in artificial environments. Here, we provide results of one-step and autoregressive reuse of latent actions in Franka Kitchen (Fig. 10). We observe that the model can effectively transfer latent actions across different scenes in different contexts, even in such out-of-distribution environments compared to the training dataset.

Additionally, these results suggest that the model can extract content-independent structures from artificial environments, and we believe it has the potential to perform well in generation and reuse tasks with an improved decoder.

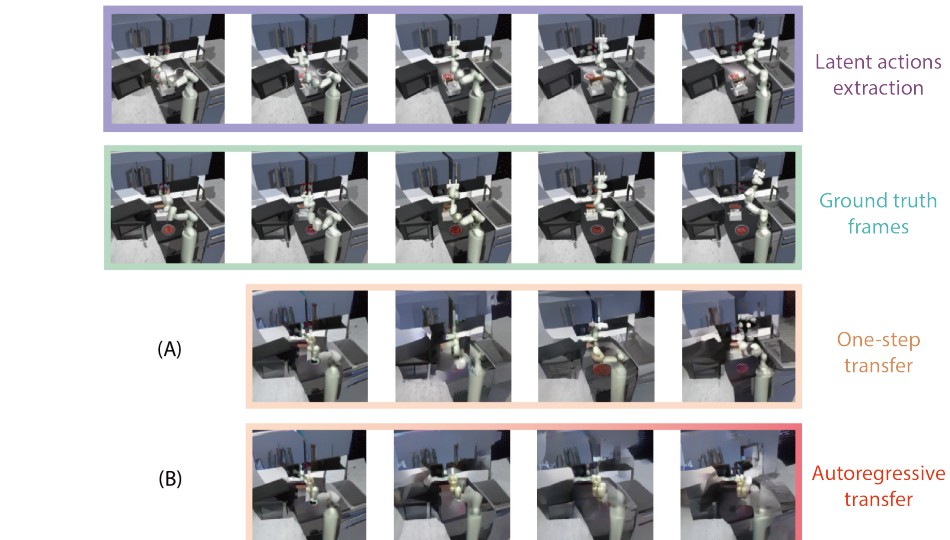

Figure 10: **Latent action transfer in artificial environments.** (A) One-step prediction by transferring the sequential latent actions in Franka Kitchen. (B) Autoregressive prediction by transferring the sequential latent actions in Franka Kitchen.

### F.3 VISUALIZATION OF BASELINE COMPARISON

We provide additional visualization of one-step prediction using LAPA, Moto, AdaWorld(LAM), and our model. The results are shown in Fig. 11. LAPA optimizes directly at the pixel level, resulting in more reliable generation of local details. In contrast, our model is optimized in the latent representation space, enabling it to better preserve overall generation quality even under large action-induced variations. In such scenarios, LAPA's generation quality tends to deteriorate, whereas our model remains robust. Moto and AdaWorld(LAM) all fail to generalize at the Franka Kitchen dataset.

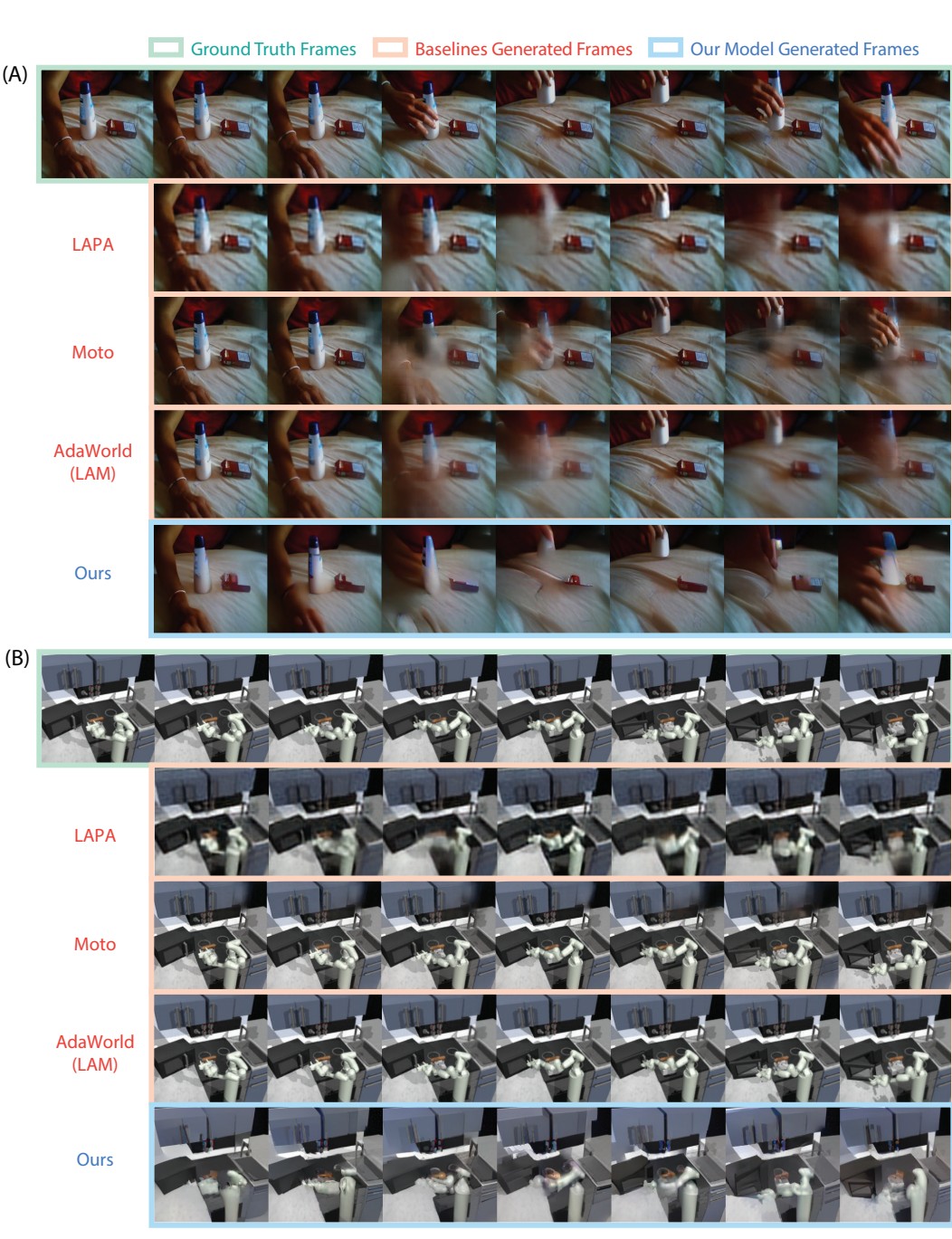

Figure 11: **Comparison of generation quality between baselines and our model.** (A) Visualization of one-step prediction on the SSV2 dataset. (B) Visualization of one-step prediction on the Franka Kitchen dataset.

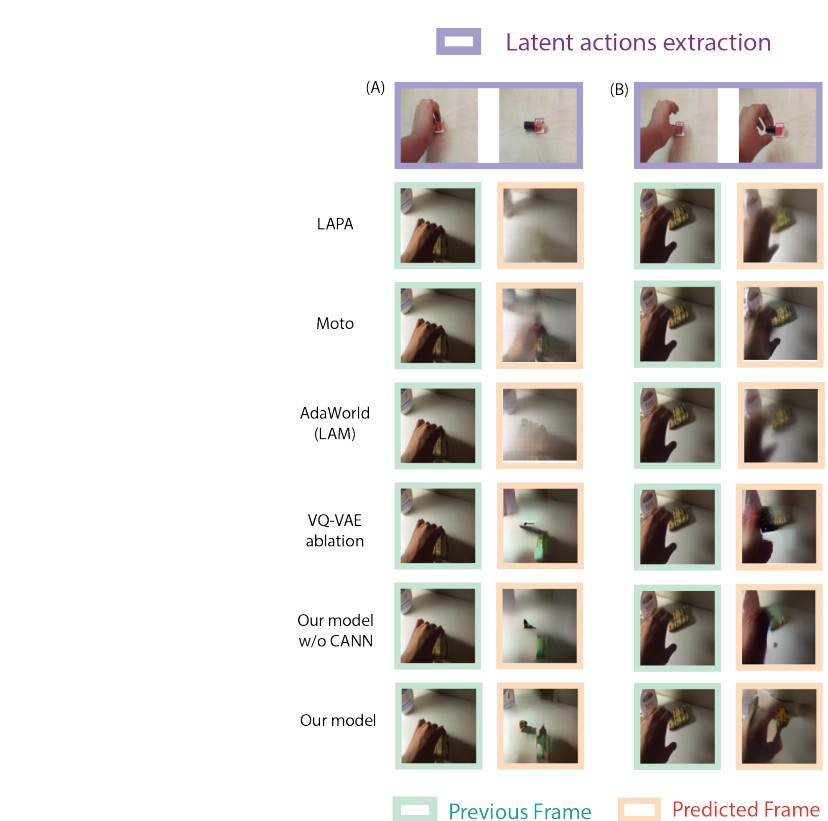

Figure 12: **Comparison of latent action transfer between baselines and our model.** One-step latent action transfer across different scenes on SSv2, using the same examples as in Fig. 4(A).

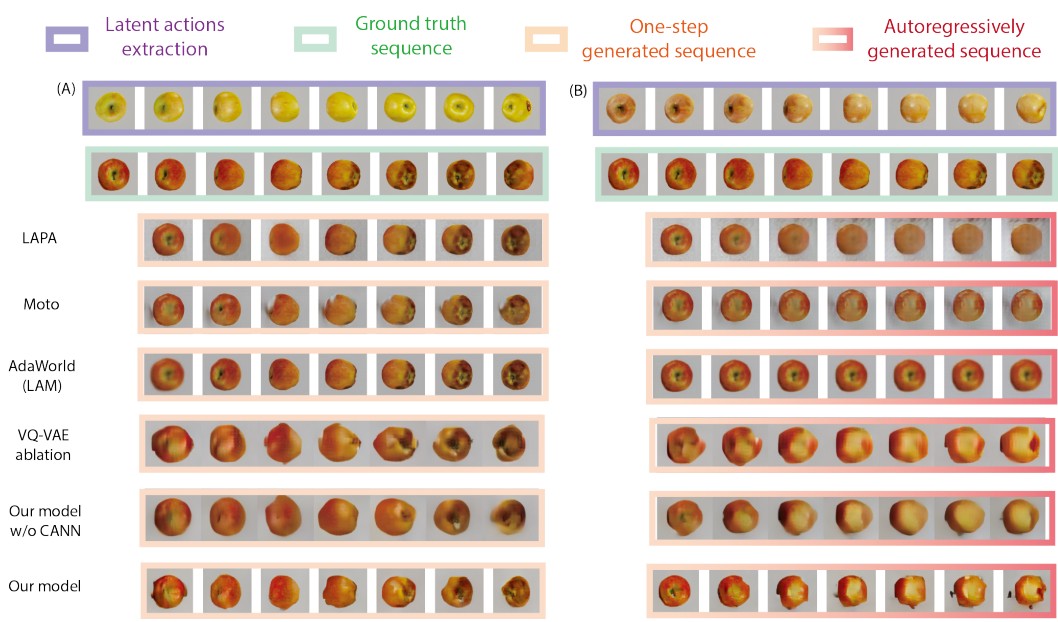

Figure 13: **Comparison of latent action transfer between baselines and our model.** One-step & autoregressive prediction by transferring the sequential latent actions, using the same examples as in Fig. 4(B, C).

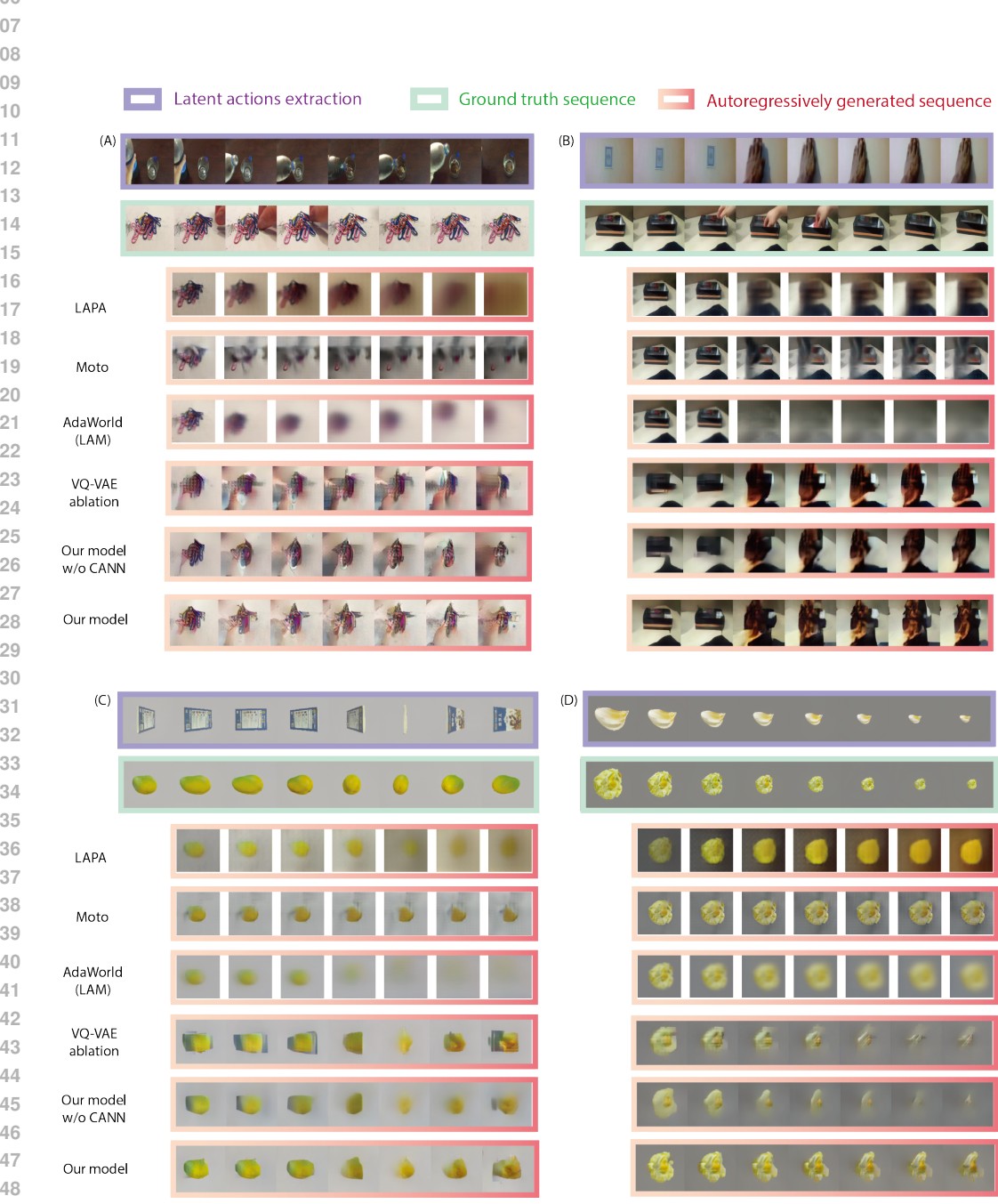

Figure 14: **Comparison of latent action transfer between baselines and our model.** (A)(B)Autoregressive reuse of latent actions on SSv2, using the same examples as in Fig. 4(D, E). (C)(D)Autoregressive reuse of latent actions on rotation and scaling dynamics across object categories, using the same examples as in Fig. 4(F, G).

# G   LEARNED LATENT SPACE DETAILS

## G.1   DIMENSIONALITY REDUCTION EXPERIMENT

In Section 5 Fig. 5(A, D), each UMAP visualization shows the embeddings of a single object. In Fig. 5(B), each UMAP figure includes embeddings of 10 pumpkins, 7 red apples, and 3 yellow apples. In Fig. 5(A, B, D), the action is fixed at $5°$ per step. The objects rotate clockwise around the vertical axis; the sequences in (A) complete two full rotations, while those in (B, D) complete one full rotation each.

## G.2   CATEGORY CLASSIFICATION DECODER

In the decoder in-class structural sharing experiment illustrated in Fig. 5(C), we construct a subset of 500 objects from OmniRotation by randomly selecting 50 categories and then randomly sampling 10 objects from each category. Then we repeat the experiment five times using this subset. In each run, we split the objects in each category into 80% for training and 20% for testing, ensuring that no object appears in both sets. The training and test samples are the per-timestep embeddings extracted from sequences of these training and testing objects.

Each object is used to construct one rotation sequence. Specifically, we initialize the object at a random view and apply a fixed action of $5°$ per step, meaning the object is rotated clockwise around the vertical axis by $5°$ at each timestep, until a full $360°$ rotation is completed. This results in a 72-frame sequence per object. The sequence is passed through our Hippocampal-Entorhinal-Inspired Coupling Model to extract the $\mathbf{p}$ and $\mathbf{g}$ embeddings, which serve as inputs for training the decoder.

The decoder is adapted from the simple latent action decoder used in Ye et al. (2024). It is an MLP consisting of two hidden layers with 128 units and ReLU activations. It is trained using the AdamW optimizer with PyTorch's default parameters. We use cross-entropy loss, a batch size of 128, and train for 30 epochs.

Fig. 5(C) reports the average training and test accuracy over the five runs, with the shaded area indicating the standard error.

## G.3   ACTION COMPOSITION ANALYSIS

We also provide an action composition analysis here. We quantitatively test for action compositionality by decomposing a diagonal movement ("move right-down 45°") into the sum of its constituent horizontal and vertical actions. Using the pumpkins and apples dataset (from Fig. 4(B)), we compare frames generated from the original diagonal latent action with those generated from the vector sum of the horizontal and vertical latent action embeddings. Fig. 15 shows that frames driven by compositional latent actions produce reasonable results comparable to real latent actions. The results in Table 6 demonstrate strong visual and quantitative similarity:

Table 6: Action composition results comparing real latent actions with composed latent actions

| ACTION TYPE | SSIM ↑ | LPIPS ↓ |
|---|---|---|
| Real latent action | 0.946±0.021 | 0.076±0.032 |
| Latent action $A + B$ | 0.944±0.023 | 0.073±0.032 |

The metrics are statistically comparable (t-test: SSIM p=0.417, LPIPS p=0.402, n=160), providing strong evidence that our latent action space supports linear composition through its path integration dynamics.

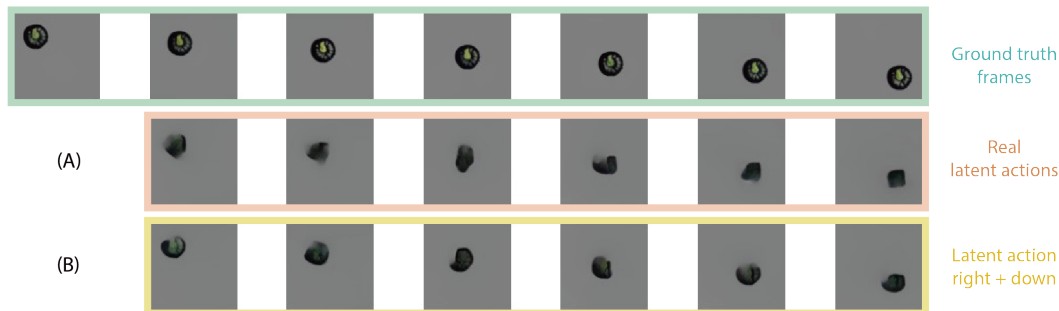

Ground truth
frames

(A) — Real
latent actions

(B) — Latent action
right + down

Figure 15: **Action composition results.** (A) One-step prediction frames driven by real latent actions. (B) One-step prediction frames driven by compositional latent actions obtained through the summation of rightward and downward latent actions extracted from the corresponding sequences.

## H   SOCIETAL IMPACT

While our foundational research can benefit embodied AI, the ability to manipulate motion dynamics carries risks. These include the potential for malicious content generation and the possibility that the model could amplify societal biases from its training data, leading to unsafe or unfair behavior.

