# OpenReview forum: "Learning and Reusing Abstract Latent Actions in a Hippocampal-Entorhinal-Inspired World Model"
_ICLR.cc/2026/Conference — Submitted to ICLR 2026_

### Official Review · Reviewer_kySr · 2025-11-01

**Soundness:** 3
**Presentation:** 4
**Contribution:** 3
**Rating:** 8
**Confidence:** 4

**Summary:**

This paper introduces a novel architecture of biological implementation of world models, inspired by neuroscience technology: HPC-MEC circuits -- Hippocampus (HPC) binds content-specific information from contents and Medial Entorhinal Cortex (MEC) extracts abstract structures among them. The proposed architecture consists of two main components: HPC-MEC coupling model and Inverse Model. HPC-MEC coupling model has a hierarchical encoder-decoder architecture and mainly works for extracting low-level dimensional MEC embeddings, so as to interact with latent actions using CANN. On the other hand, inverse model extracts latent actions between the given two MEC embeddings. As experimental results, this approach generally outperforms other strong baselines on 7 different tasks, along with OOD generalization tasks. Interestingly, authors also provide latent space analysis on rotated object tasks, which implies the model's feature shapes abstract structures while in-class differentiation is formed.

**Strengths:**

1. **Strong Novelty.** Drawing from HPC-MEC circuit behavior observed in neuroscience to implement a biological world model represents a solid research contribution. The connection is not forced, and the motivation is well-justified. The approach also offers interesting points of departure for future work.

2. **Clear Illustration and Explanation.** Figures 1 and 2 effectively present the overall approach in a comprehensive manner. The figure quality is excellent and shows a polished presentation. Furthermore, even when explaining complex architectural details, the paper maintains clarity through step-by-step descriptions of the actual architecture's components. The explanations are detailed but accessible, making it easy to follow the proposed framework and related work.

3. **Comprehensive Experiment Design.** The evaluation uses three different categories and various objectives across 7 different datasets, providing substantial evidence of the model's behavior. Notably, the rotation task and OOD task evaluations, which specifically leverage the HPC-MEC circuit properties, demonstrate significant advantages and offer valuable insights for future research.

**Weaknesses:**

- **Downstream Tasks with Learned World Models.** The paper only evaluates the quality of generated video output compared to ground truth video, which may not sufficiently demonstrate the model's ability to capture world physics. While this is acceptable given the paper's focus on more fundamental problems, including a few downstream tasks (e.g., robotic task performance using this world model) could provide stronger evidence of the advantages that the HPC-MEC module offers for robotics applications.

**Questions:**

- The latent space analysis was quite impressive. To clarify, in Figure 5-B, g^{gen} shows overlapped representations between two colored apples, but p^{gen} successfully differentiates them. How can this behavior be explained?

---

> ### Author Response · Authors · 2025-11-19
> **Rebuttal by authors**
>
> Thank you for your high evaluation and for recognizing the novelty and solidity of our research contribution. We appreciate your positive feedback on the clarity of our architectural explanations and figures, as well as your acknowledgment of the substantial evidence provided by our extensive experiments on diverse datasets.
>
> **Responses to Weakness:**
>
> Thank you for this suggestion. As you said, visual prediction and zero-shot action transfer are fundamental tasks for validating the _disentanglement_ and _abstraction_ capabilities of these representations. We strongly agree that evaluating our HPC-MEC world model on robotic tasks is an important direction. A common approach is to decode latent actions into ground-truth actions for planning. The model's performance on such downstream tasks is directly correlated with its ability to decode actions accurately. Therefore, as a quantitative proxy for this capability, we supplemented our paper with an experiment on **decoding the action type from our learned latents**.
>
> We used the OmniObject3D dataset to create sequences with varied transformations (rotation, horizontal/vertical translation, scaling). In each sequence, the object's category, size, and position were randomized. We fed these sequences to our model for zero-shot generalization and used the resulting latents to train an action decoder. The results are shown in Table A.
>
> Table A
>
> | Embedding | $ p^{inf} $ | $ g^{inf} $ | $ \Delta p^{inf} $ | $ \Delta g^{inf} $ | $ z $  |
> | --- | --- | --- | --- | --- | --- |
> | Action Decoding Accuracy↑ | $0.3330 \pm 0.0163$ | $0.3486 \pm 0.0156$ | $0.8386 \pm 0.0263$ | $0.8868 \pm 0.0212$ | $\mathbf{0.9064 \pm 0.0145}$ |
>
>
> We found that $ z $  decodes latent actions with the highest accuracy, followed by $ \Delta g $. This quantitatively demonstrates that our model's latents effectively capture the action dynamics.
>
> Furthermore, we are actively exploring this in our follow-up work by using our model for **goal-directed planning with Model Predictive Control (MPC)**. Our hierarchical model is ideal for this because the content/structure separation also provides a natural **stable vs. plannable representation separation**:
>
> + **Stable representations (HPC)** tell the agent "where I am" by encoding contextual details.
> + **Plannable representations (MEC)** act as a "latent compass," telling the agent "the direction I should go" in the latent space.
>
> Because the MEC isolates the abstract structure, the direction calculated in this space is more robust. We can then accelerate the MPC sampling process using this directional information from the MEC and use the chosen latent actions for imagination (rollouts). However, a full implementation of this is beyond the scope of our current contribution.
>
> **Responses to Question:**
>
> Thank you, that's an excellent observation, and it highlights the central mechanism of our model.
>
> The HPC embedding is designed to be a **joint encoding of both content and structure**. During training, the $p^{\text{gen}}$ (which is decoded from $g^{\text{gen}}$) is forced to **align** with $p^{\text{inf}}$ (which is encoded from the input and retains content-specific details). This alignment objective forces the decoder to implicitly learn "content binding." The mechanism is analogous to an autoencoder: The latent representation ($g^{\text{gen}}$) is a **bottleneck** and may lose fine-grained details. However, the decoder (the $ g \rightarrow p$ pathway) is trained to **reconstruct** the original, richer representation ($p^{\text{inf}}$) from that bottleneck.
>
> Therefore, content information like "red" and "yellow" is "re-encoded" by the decoder onto $p^{\text{gen}}$. This is precisely why, in the UMAP visualization, the trajectories for the different colored apples are clearly separated in the $p^{\text{gen}}$ space.

---

> > ### Comment · Reviewer_kySr · 2025-11-27
> >
> > I appreciate the authors' considerable response to my concerns.
> >
> > **Regarding the authors' response to weakness**, I'm grateful to observe strong potential towards downstream tasks, such as embodied and robotics tasks, through the provided proxy downstream task results showing approximately 90% accuracy on the action prediction task, and the detailed scenarios and implementation plans shared for potential future work.
> >
> > **Regarding the response to my question**, my concerns are fully resolved. Thanks for the authors' clear explanation!
> >
> > I will maintain my score. The paper deserves this high score.

---

> > > ### Author Response · Authors · 2025-11-27
> > >
> > > We sincerely appreciate your encouraging feedback and the time you invested in reviewing our paper. We are pleased that you found the additional results on action decoding valuable, and we will incorporate the supplementary experiments and detailed explanations into the revised version. Thank you again for your continued support and for maintaining your positive assessment!

---

### Official Review · Reviewer_Fvm6 · 2025-11-01

**Soundness:** 2
**Presentation:** 2
**Contribution:** 2
**Rating:** 2
**Confidence:** 4

**Summary:**

The method learns an action representation by adding additional layers in the bottleneck of VAR [1], a frozen autoregressive video VQ-VAE. The method uses the encoder to obtain observation embeddings, then infers latent actions via an inverse model and applies them within a hierarchical Continuous Attractor Neural Network (CANN) structure for path integration. The model is evaluated on Something-Something v2, COIL-100, and simulated OOD benchmarks both qualitatively and in terms of SSIM and LPIPS against three world modeling baselines.

[1] Visual Autoregressive Modeling: Scalable Image Generation via Next-Scale Prediction

**Strengths:**

1. The method avoids any action labels and builds the latent-action space through inverse modeling in the bottleneck of VAR.

2. Evaluations are on real and simulated datasets and the authors motivate the method by action transfer across visual domains.

3. The motivation and method are motivated by biology (HPC-MEC circuit) and qualitative evidence for emergence of separation between HPC and MEC in terms of content features and geometric features is provided (Fig 5).

**Weaknesses:**

1. My main concern is that the frozen VQ-VAE backbone differs from those used in LAPA, Moto, or AdaWorld but no comparison is shown to the pretrained autoregressive VAR model itself [1]. This makes it unclear whether performance improvements and action stem from the new architecture or from feature quality of the original VAR.

2. Following my point above, the hierarchical design (transformers + CANN module) is heavy yet yields modest practical advantages. There is no ablation to show how the pretrained VQ-VAE performs without these components. Furthermore, it is unclear whether the CANN-based path integration outperforms a simpler architecture (e.g. MLP) of equal capacity when trained under identical conditions.

3. The qualitative evaluation results do not show robustness or realism for all types of actions (e.g. Fig. 4 D and E). Furthermore, the autoregressive generated trajectories seem to be prone to compounding error (e.g. Fig 4 C).

4. Although the method claims to learn latent action representations, no metric is provided for decoding actions from the learned latents. Perceptual similarity metrics are not a suitable proxy for direct action decoding.

5. The neuroscience background is verbose and disconnected from the rest of the paper.

[1] Visual Autoregressive Modeling: Scalable Image Generation via Next-Scale Prediction

**Questions:**

1. How does the model compare directly to the underlying VAR without adding HPC-MEC layers?

2. Can you provide quantitative disentanglement metrics (e.g., mutual information between latent actions and pixel-space transformations or decoding actions from HPC vs MEC tokens) rather than qualitative UMAPs?

3. Does fine-tuning the backbone and decoder jointly while learning the HPC-MEC layers improve performance?

4. Can you provide compute and wall-clock time comparison between the method and the baselines (including original VAR)?

5. Does the model enable any improvement in downstream RL and control tasks relative to baselines like LAPA?

---

> ### Author Response · Authors · 2025-11-19
> **Rebuttal by authors (Part 1/3)**
>
> Thank you for your valuable and constructive feedback. We are grateful for your recognition of our self-supervised approach to building a latent action space without action labels. We also appreciate your acknowledgement of our demonstration of action transfer across diverse domains, and our qualitative results that show the emergent separation of content and structure.
>
> **Weakness 1/2 & Question 1: Ablation on pretrained VQ-VAE**
>
> Thank you for this crucial question. We would like to clarify our motivation for adopting the pretrained VQ-VAE and demonstrate that the observed gains stem from our novel architecture rather than the VQ-VAE.
>
> Our goal is to avoid the common **trade-off between pixel-level generation quality and latent action abstraction**[1], which is common in pixel-level generation models such as LAPA, Moto, and AdaWorld (LAM). Inspired by JEPA, we shift from pixel reconstruction to **prediction in a structured latent space**. We choose the VQ-VAE used in the VAR model because it provides a compressed representation (16×16×32), significantly more compact than Dinov2/MAE (16×16×768) and raw images (256×256×3).
>
> While the VQ-VAE provides high-quality features, it is not the key component responsible for learning abstract actions. Our ablation studies support this. All ablations in Table 2 used the _same_ VQ-VAE encoder, yet both the “unified space” model (without HPC–MEC disentanglement) and the “w/o CANN” model (without path integration) failed the latent action reuse task ($R$ metric) and produced lower generation quality. This shows our architecture provides key functions beyond the encoder.
>
> To further support our claim, we ran the new ablation you suggested: **a VQ-VAE combining an inverse model and an MLP-based forward model** (since a standalone VQ-VAE cannot learn latent actions). The inverse model takes two consecutive embeddings from the VQ-VAE to generate a latent action $z_t$, which is concatenated with the previous embedding to predict the next using the MLP. The results shown in Table A confirm this model's latent actions have poor transferability and abstraction ($R$ metric).
>
> Table A
>
> | Dataset | Model | SSIM ↑ (one-step) | SSIM ↑ (autoregression) | LPIPS ↓ (one-step) | LPIPS ↓ (autoregression) | R ↑ (one-step) | R ↑(autoregression) |
> | :--- | :--- | :--- | :--- | :--- | :--- | :---: | :---: |
> | SSV2 | VQ-VAE ablation | $0.717\pm0.023$ | $0.677\pm0.031$ | $0.313\pm0.013$ | $0.378\pm0.017$ | | |
> | | Our model | $\mathbf{0.752\pm0.019}$ | $\mathbf{0.687\pm0.018}$ | $\mathbf{0.274\pm0.026}$ | $\mathbf{0.356\pm0.015}$ | | |
> | Franka Kitchen* | VQ-VAE ablation | $0.576\pm0.037$ | $0.523\pm0.038$ | $0.385\pm0.018$ | $0.445\pm0.022$ | | |
> | | Our model | $\mathbf{0.705\pm0.004}$ | $\mathbf{0.551\pm0.005}$ | $\mathbf{0.253\pm0.003}$ | $\mathbf{0.426\pm0.003}$ | | |
> | OmniObjeet3D action transfer * | VQ-VAE ablation | $0.892\pm0.009$ | $0.883\pm0.009$ | $0.158\pm0.009$ | $0.177\pm0.009$ | $2.035\pm0.229$ | $1.796\pm0.173$ |
> | | Our model | $\mathbf{0.902\pm0.010}$ | $\mathbf{0.891\pm0.009}$ | $\mathbf{0.120\pm0.008}$ | $\mathbf{0.156\pm0.008}$ | $\mathbf{3.201\pm0.435}$ | $\mathbf{2.482\pm0.460}$ |
>
> > \* indicates out-of-distribution datasets.
>
> This failure stemmed from two factors: (1) the inverse model's input was the content-rich observation embedding, not the structure-only MEC embedding, and (2) the forward model lacked the inductive bias of the CANN module.
>
> We also ablated the VQ-VAE entirely to perform pixel-level prediction. We found this required increasing our HPC, MEC, and latent action dimensions by at least **16-fold** just to match the original model's generation quality, which in turn sacrificed the abstraction of the latent actions.
>
> In conclusion, these experiments confirm that the VQ-VAE alone is insufficient. Successful abstraction requires _both_ the input **disentanglement** (provided by our HPC-MEC architecture) and the appropriate **model bias** (from CANN-based path integration). We hope these experiments could address your concerns. Please let us know if you have any further doubts.
>
> [1] AdaWorld: Learning Adaptable World Models with Latent Actions, ICML 2025
>
> **Weakness 2: Ablation on CANN module**
>
> The CANN-based ablation you mentioned is indeed included in **Table 2** as “**Our model w/o CANN**”. In that experiment, we replaced the CANN component with an MLP of equivalent parameters, where $ g^{inf}_t $ and $ z_t $ were concatenated as input to the MLP to predict $ g^{gen}_t $. The results showed it was inferior to the CANN version in both generation quality and disentanglement during transfer. In the revised version, we will make the ablation settings explicitly clear.

---

> ### Author Response · Authors · 2025-11-19
> **Rebuttal by authors (Part 2/3)**
>
> **Weakness 3: Realism of all types of actions & compounding error**
>
> Thank you for this observation. We realize that our original description may not have made our intentions fully clear. Here, we clarify the capabilities demonstrated in Figures 4(D, E) and outline how we plan to address the autoregressive compounding error.
>
> + **Regarding Robustness or Realism of Actions:**
>     - Figures 4(D, E) face a more challenging test: transferring dynamics between sequences with _dissimilar_ objects and motions. Critically, while the new sequence inherits the _motion_ from the source, it **does not leak source textures** (e.g., the original bottle's reflection or the hand's color). This provides strong evidence that our latent actions are successfully disentangled and content-independent. We agree that the fine-grained detail generation is not yet perfect, and we plan to address this in future work by exploring better fusion methods, such as FiLM.
> + **Regarding Compounding Error in Figure 4(C):**
>     - The autoregressive generation in Figure 4(C) faces a specific challenge: the first frame (the red apple) lacks information about the object's unseen back. Our model reasonably predicts a light-yellow color to fill in this missing information.
>     - While all autoregressive models inherently face compounding error, our model **quantitatively outperforms all baselines** (LAPA, Moto, AdaWorld) in the autoregressive setting. We also note that this error accumulation is inherent to path integration and is biologically plausible (MEC grid cells also accumulate errors).
>     - For future work, we are exploring **Lie group operators** for latent action modeling. Preliminary results suggest that Lie groups' structural properties (invertibility, orthogonal operators) can reduce compounding errors. Since rotation operators are mathematically equivalent to CANN dynamics, this integrates naturally with our framework. We plan to demonstrate these improvements in follow-up work.
>
> **Weakness 4 & Question 2: Metric for decoding actions**
>
> Thank you for this great suggestion. We have added your recommended action-decoding experiment, which further validates the effectiveness of our learned latent representations.
>
> We constructed a new dataset using OmniObject3D with sequences containing different transformations (rotation, horizontal/vertical translation, and scaling). Each sequence features an object where the category, initial position, orientation, and size are randomized. We fed these sequences to our model for zero-shot generalization and used the resulting latents to decode the action. We trained action decoders of the same hidden size to decode the **action type**, measuring the abstraction and action semantics of the latent representations. The results are shown in Table B.
>
> Table B
>
> | Embedding | $p^{inf}$ | $g^{inf}$ | $\Delta p^{inf}$ | $\Delta g^{inf}$ | $z$ |
> | --- | --- | --- | --- | --- | --- |
> | Action Decoding Accuracy ↑ | $0.3330 \pm 0.0163$ | $0.3486 \pm 0.0156$ | $0.8386 \pm 0.0263$ | $0.8868 \pm 0.0212$ | $\mathbf{0.9064 \pm 0.0145}$ |
>
> We found that $z$ achieves the highest accuracy for decoding latent actions, followed by $\Delta g^{inf}$. This strongly demonstrates that our model's learned latents are effective and semantically meaningful.
>
> **Weakness 5: Verbose Neuroscience Background**
>
> Thank you for this valuable feedback. We will revise the neuroscience background to more clearly demonstrate its direct connection to our computational approach.
>
> The goal of our work is not only to propose a disentangled world model for learning abstract latent actions, but also to establish a functional correspondence between the brain's HPC-MEC circuit and AI's world models. This correspondence extends beyond just spatial tasks. We argue that the way grid cells encode non-spatial transitions[2-4] is functionally analogous to the _latent actions_ in modern world models — **a connection that has been largely underexplored**.
>
> Moreover, neuroscience is not just an analogy; it provides a specific **inductive bias**. The HPC-MEC circuit naturally introduces the disentanglement of content and structure. By using the CANN dynamics, we demonstrate that this bio-inspired approach may provide a semantically meaningful alternative to tokenized or discrete latent actions (As pointed out by Reviewer fVRp).
>
> While our model and experiments are AI-focused, the background is essential for bridging these two fields and allowing researchers in both AI and neuroscience to see this functional link. We will revise the abstract and introduction to make this correspondence more explicit and accessible to an AI-focused audience.
>
> [2] Map making: Constructing, combining, and inferring on abstract cognitive maps. Neuron 2020
>
> [3] Hexadirectional coding of visual space in human entorhinal cortex. Nature Neuroscience, 2018
>
> [4] Grid-like neural representations support olfactory navigation of a two-dimensional odor space. Neuron, 2019

---

> > ### Author Response · Authors · 2025-11-19
> > **Rebuttal by authors (Part 3/3)**
> >
> > **Question 3: Jointly finetune with VQ-VAE**
> >
> > We believe this would likely be detrimental to performance. If we understand correctly, joint fine-tuning would require adding a pixel-level reconstruction loss to pass gradients back to the VQ-VAE backbone. This introduces two significant problems:
> >
> > + **Loss of Generalization:** Full fine-tuning of the pretrained VQ-VAE on a single dataset like SSv2 would likely cause it to lose its powerful generalization capabilities and could lead to representation collapse.
> > + **Re-introducing the Trade-off:** More importantly, the gradients from this pixel-level loss would backpropagate to the latent action $z$. This would reintroduce the very **trade-off between abstraction and generation quality** that our latent-space-only approach was designed to avoid. The model would be forced to compromise the purity of its abstract dynamics to fix small pixel errors, which would directly harm the learning of abstract latent actions. The experiment without VQ-VAE, as shown in **Weakness 1** can also verify this statement.
> >
> > **Question 4: Compute and wall-clock time comparison**
> >
> > We ran a new set of experiments on a single NVIDIA A100 GPU to fairly compare the inference throughput. We used a consistent batch size of 16 and a sequence length of 8, averaging the results over 100 batches to calculate the **Inference FPS** (higher is better) and **Average Time per Batch** (lower is better).
> >
> > Table C
> >
> > | Model | VQ-VAE (reconstruction only) | LAPA | Moto | AdaWorld(LAM) | Ours(Full Model) |
> > | :---: | :---: | :---: | :---: | :---: | :---: |
> > | Inference FPS | 86.44 | 205.33 | 55.22 | 35.60 | 84.00 |
> > | Average time per batch (s) | 1.481 | 0.623 | 2.318 | 3.595 | 1.523 |
> >
> > Our **HPC-MEC module adds almost no computational overhead**. The reason for this high efficiency is that our module operates _entirely in the latent space_ between the VQ-VAE encoder and decoder. Also, our full model's inference speed is faster than Moto and AdaWorld(LAM). We believe this analysis shows that our model's advantages are achieved without incurring an unreasonable computational penalty. We will add this to the appendix, and thank you for prompting this valuable addition.
> >
> > **Question 5: Improvement on RL or control tasks**
> >
> > While a common downstream task is using latent actions as a proxy for ground-truth actions in VLA models, we believe our model's primary strength lies in **planning and imagination** in latent space. We are actively exploring the use of this HPC-MEC world model with **Model Predictive Control (MPC)** for goal-directed planning tasks (relative to the AdaWorld baseline). Our hierarchical model is uniquely suited for this because its separation of content (HPC) and structure (MEC) also naturally facilitates a separation of **stable vs. plannable representations**:
> >
> > + The **stable representations in HPC** (content-specific details) tell the agent "where I am" in a specific context.
> > + The **plannable representations in MEC** (abstract structures) tell the agent "the direction I should go" in the latent space.
> >
> > We can then leverage this separation to accelerate the MPC sampling process using the directional information from the MEC, while using the learned latent actions to perform imagination (i.e., autoregressive rollouts). This is a key advantage of our neuro-inspired design, and we will leverage this ability in our future work.

---

> ### Comment · Reviewer_Fvm6 · 2025-11-27
> **Response to rebuttal**
>
> Thank you for your response. Two of my concerns have (W1 and W2) have been addressed.
>
>
> However, W3 and W4 and Q3 and Q5 have not been properly addressed:
>
>
> W3: Please provide the same visualizations of Figure 4 (predictions and trajectories for same data samples) for AdaWorld, Moto, LAPA, vanilla VAR, and your MLP-based model (w/o CANN). The model should offer better qualitative generation if the contribution is claimed to be learning structured action representations.
>
>
> W4: Results without compared baselines are meaningless. Provide the same metrics with a similar evaluation protocol for AdaWorld, Moto, LAPA, vanilla VAR, and the MLP-based model (w/o CANN). Also, clarify the evaluation protocol for action decoding; do you feed the frozen concatenated representations of $x_t$ and $x_{t+1}$ to an MLP to decode the action (i.e. common practice in equivariant SSL [1, 2, 3])? If so, state the protocol and refer to the literature.
>
>
> Q3: The argument is not convincing. Yes, I meant fine-tuning the whole VQ-VAE but only for few-shot adaptation of the VAE to a given downstream task (transferred action space) which can possibly improve the fidelity of generated outputs (e.g. Fig 4).
>
>
> Q5: Action representation baselines (AdaWorld, Moto, and LAPA) all report results on planning and control benchmarks which is lacking in the present work. Arguably, planning is the most important application of world models that have good action modeling, yet no results are reported regarding that. There should be at least one such a benchmark in the paper to show its competitiveness with the baselines. Furthermore, the model can be compared with V-JEPA 2 [4] (not required now but should be mentioned in the paper) by training the CANN circuit on downstream robotic data and then performing MPC-based planning (although I think the generative nature of the model would make MPC-style planning very inefficient similar to generative baselines in [4])
>
>
> [1] Self-supervised learning of Split Invariant Equivariant representations
>
> [2] In-Context Symmetries: Self-Supervised Learning through Contextual World Models
>
> [3] seq-JEPA: Autoregressive Predictive Learning of Invariant-Equivariant World Models
>
> [4] V-JEPA 2: Self-Supervised Video Models Enable Understanding, Prediction and Planning

---

> ### Author Response · Authors · 2025-11-28
>
> We sincerely appreciate your feedback. We have promptly conducted the additional experiments you mentioned and will now address the results point-by-point.
>
> **W3:** Thank you for your comments. As suggested, we have added visualizations for all the mentioned baselines and ablations in Appendix F.3 (in our updated PDF). The results show that the baselines struggle with effective action transfer, limited by their latent action representation and generation quality. In contrast, the ablations exhibit clear content leakage, which is particularly evident in the rotation transfer of different object categories, along with a degradation in output quality. Taken together with our quantitative results (Tables 1 & 2), these findings robustly demonstrate that our model achieves superior latent action learning and higher generation quality compared to other latent action models.
>
> **W4:** Thanks for your follow-up question. We provide all the action decoding accuracy of baselines and our ablations. The result is shown in Table D. Our latent actions outperform all other baselines and ablations.
> Table D
> | Embedding |  $z$ | AdaWorld(LAM) | LAPA | Moto | VQ-VAE ablation $z$ | Our model w/o CANN $z$|
> |  --- |  --- |  --- |  --- |  --- |  --- | --- |
> | Action Decoding Accuracy ↑ | $\mathbf{0.9064 \pm 0.0145}$ | $0.6395 \pm 0.0192$ | $0.8259 \pm 0.0155$ | $0.7190 \pm 0.0120$ | $0.8523 \pm 0.0247$ | $0.8768 \pm 0.0117$ |
>
> We use the action decoding paradigm commonly used in latent action decoding[1]. We do not feed the frozen concatenated representations of $x_t$ and $x_{t+1}$ to an MLP to decode the action. Instead, we first use our trained model to extract the latent action representation $z$ from the video sequence, and then feed the latent action $ z_t$ directly into the action decoder (MLP) to decode the action $a_t$, which represents the type of the current transition on time $t$. For each embedding setting, we train a latent action decoder following LAPO [1]. Each decoder is implemented as a fully connected network with hidden sizes of 128 and 128. For $\Delta g^{inf}$and $\Delta p^{inf}$ (in Table B), we directly calculate the difference from the MEC and HPC embeddings and feed them to the MLP decoder to decode action types. We will add this evaluation protocol to our revised version to make it clear.
>
> [1] LEARNING TO ACT WITHOUT ACTIONS, ICLR 2024
>
> **Q3:** We appreciate the clarification and agree that fine-tuning the VQ-VAE on downstream datasets would indeed improve the visual fidelity of the generated frames. However, we intentionally kept the VQ-VAE frozen to rigorously evaluate the **structural generalization** of our learned dynamics, rather than the adaptation capability of the visual encoder. By treating the VQ-VAE as a fixed, foundational visual frontend (analogous to discriminative models using frozen DINO features), we ensure that the successful control of novel objects stems from the world model's ability to reuse abstract latent actions, rather than the encoder overfitting to the target textures. This setup serves as a stricter stress-test of our core contribution: a disentangled world model capable of zero-shot structural transfer.
>
> **Q5:** We appreciate the reviewer’s emphasis on planning and will explicitly discuss V-JEPA 2 and the efficiency of performing MPC within our low-dimensional latent space in the revision. However, we respectfully highlight that our primary contribution is **structural generalization**: while baselines like LAPA and Moto report planning results, our experiments demonstrate that they fail to predict accurate dynamics when shifted out of distribution (as shown in Table 1 and Figure 11 ), whereas our model maintains robust performance. We argue that high-fidelity dynamics prediction on control benchmarks (e.g., Franka Kitchen) serves as a rigorous and necessary proxy for world model quality, verifying that the model learns a transferable "map" of dynamics before optimizing a specific "navigator" or policy.
>
> Furthermore, the baseline methods utilize significantly enhanced planning paradigms (e.g., VLMs in LAPA/Moto to form VLA for planning, and the pretrained OpenSora as a world model in AdaWorld), making a direct planning comparison an unbalanced assessment. To ensure a fair comparison of the latent action and world model components, MPC-based planning is the most suitable benchmark, as it explicitly requires a world model (unlike behavior cloning). We are actively conducting these experiments and fine-tuning all LAMs in baseline models on the Robosuite task for MPC-based planning. While the full results may not be available before the rebuttal deadline, we would like to emphasize that planning performance in this setting is highly correlated with the quality of latent actions and generative fidelity, which are strongly demonstrated in our Table 1 and action decoding studies. We are committed to including these planning results in the revised manuscript.

---

### Official Review · Reviewer_T6rd · 2025-11-04

**Soundness:** 3
**Presentation:** 3
**Contribution:** 4
**Rating:** 8
**Confidence:** 4

**Summary:**

The paper proposes a cognitive-/neuro-science-inspired artificial neural network architecture that implements a disentangled world model that learns abstract latent actions to predict next-state observations purely from video frames in a self-supervised manner.

The architecture uses a pretrained frozen vector-quantized autoencoder to generate initial frame embeddings, which are then processed by a stack of temporal causal transformers with dimensionality reduction in the latent codes. A Continuous Attractor Neural Network (CANN) predicts the delta to the next latent state given a latent action state and is regularized to be close to this delta. The latent action comes from an action model that generates this latent action code given two consecutive latent codes.

**Strengths:**

- Reproducibility: The provided source code is structured and easy to understand

 - Learning of visually independent latent actions

 - Strong autoregressive performance

 - Ablations show the relevance of the used predictive CANN model

**Weaknesses:**

- The authors claim “MEC captures shared dynamics across objects, while the HPC retains object-specific information,” but as I understand, MEC and HPC are both temporal causal transformers, with HPC having a narrow bottleneck, so how is this claim justified?

- Temporal recurrence in Figure 2 is a bit misleading since a causal transformer is not really recurrent in the RNN sense, so I would rather call this temporal information flow or something in that spirit.

- The ablation models need to be explained in more detail; for example, what exactly is meant by “unified space model”?

- More details on how the CANN was ablated would be helpful. I am not fully convinced that a transformer-like architecture with the same inputs and regularizations as the CANN would perform much worse.

- The abstract is a bit hard to read and could be addressed more toward a computer science audience. Think of it more like an advertisement of your paper; currently the cognitive and neuro-science terminology makes it a bit hard to parse what the paper will be about and what the exact contributions are.

- Figure 2 “dashed green arrow” should be “dashed purple arrow.”

**Questions:**

See Weaknesses

---

> ### Author Response · Authors · 2025-11-19
> **Rebuttal by authors**
>
> We sincerely appreciate your strong support and your recognition of our work's reproducibility and technical strengths. We are glad that the visually independent latent actions and the ablations for the CANN model demonstrated the effectiveness of our approach.
>
> **Weakness 1: The role of HPC & MEC**
>
> Thank you for this question, which points to the core of our model's design. We would like to gently clarify a key architectural detail, as we believe there may have been a small typo in the review: in our model, the **MEC serves as the bottleneck, not the HPC**. As shown in Table 4, the higher-dimensional HPC embeddings ($ p $, size 8192) are compressed into the _lower-dimensional_ MEC embeddings ($ g $, size 4096). This strong information bottleneck is intentional and is the key to the disentanglement we claim. It forces the model to discard content-specific details (like object-specific textures, which the HPC retains) while preserving the abstract structural information (like rotational pose) needed for dynamics.
>
> **Weakness 2: Temporal recurrence in Figure 2**
>
> You are absolutely right, thank you for pointing this out. We will revise Figure 2 and the corresponding text to use "Temporal Dependence" instead. We appreciate the correction!
>
> **Weakness 3: Clarification on "unified space model" ablation**
>
> Thank you for the kind suggestion, and we will clarify this in our revised version. The "Our model w/ unified latent space" is an ablation where we removed the MEC layer to test the necessity of our disentangled architecture. To do this, we "grafted" the inverse model and the CANN module directly onto the single, high-dimensional HPC layer. This means:
>
> 1. The inverse model no longer inferred $ z_t $ from the $ g $ embeddings. Instead, it was forced to infer $ z_t $ from consecutive HPC embeddings ($ p_t^{inf} $ and $ p_{t+1}^{inf} $), which retain object-specific content.
> 2. The CANN module's path integration then operated directly on this content-entangled HPC state ($ p_t $).
>
> As shown in Table 2, this "unified" model failed the latent action reuse task, suffering from significant "texture leakage" (a low R metric). This result supports our central claim that the HPC-MEC separation is critical for disentanglement.
>
> **Weakness 4: Clarification on "CANN module" ablation**
>
> Here is the exact design of the CANN ablation (the "Our model w/o CANN" from Table 2). We replaced our CANN-based forward model with a standard MLP of equivalent capacity. This ablated model took the concatenation of the latent action $ z_t $ and the current state $ g_t^{gen} $ as input, and was trained to directly predict the next state $ g_{t+1}^{gen} $ (i.e., $ g_{t+1}^{gen} = \text{MLP}(\text{concat}(z_t, g_t^{gen})) $). We call this "state-to-state" prediction.
>
> As Table 2 shows, this "state-to-state" prediction performed worse. The reason, we argue, is the loss of a critical inductive bias. **Our forward Model (with CANN)** does not predict the full next state. It predicts **only the change** ($ \Delta g_t $). This is a "transition-to-transition" task: The inverse model learns $ z_t $ from a transition ($ \Delta g^{inf} $), and the forward model only uses $ z_t $ to predict a corresponding transition ($ \Delta g_t $). This "reduces the reconstruction pressure" on $ z_t $, forcing it to be purely about dynamics and thus more content-independent.
>
> **Weakness 4: Neuroscience terminology in the abstract**
>
> Thank you for this practical advice. We find that the current abstract might overemphasize the neuroscience, making it hard to parse for a CS audience. The reason we introduce the neuroscience background is to establish the _functional correspondence_ between this model and the brain's HPC-MEC circuit.  We will revise it to lead with the clear technical contributions and benefits (e.g., self-supervised learning, robust disentanglement, and OOD generalization), making the paper's value much more accessible.
>
> **Weakness 5: Typo**
>
> Thank you for your careful reading and for catching this typo. We have already corrected it in the latest version of the paper.

---

### Official Review · Reviewer_fVRp · 2025-11-10

**Soundness:** 3
**Presentation:** 2
**Contribution:** 2
**Rating:** 4
**Confidence:** 3

**Summary:**

This paper presents a self-supervised framework that can simultaneously learn latent actions and a world model directly from real-world videos. The approach is inspired from the hippocampal - entorhinal (HPC - MEC) circuit, a biological system known for encoding spatial and abstract structures in the brain. The model separates representations into content-specific (HPC) and structure-specific (MEC) components. Latent actions are learned via an inverse model operating in the MEC latent space, where the authors employ Continuous Attractor Neural Network (CANN) dynamics to represent velocity-like transformations.

The proposed model is evaluated on Something-Something V2, COIL-100, and several robotic simulation datasets (e.g., Franka Kitchen, Push-T). Results show strong generalization and transfer - latent actions extracted from one context (e.g., a hand motion or object rotation) can be reused to predict analogous dynamics in new visual scenes. Quantitative results on SSIM and LPIPS confirm better visual fidelity and lower perceptual error than baselines such as LAPA, Moto, and AdaWorld.

**Strengths:**

(+) The paper offers an elegant synthesis between computational neuroscience and self-supervised world modeling. This connection gives the model interpretability and conceptual depth beyond typical deep learning architectures.

(+) Modeling latent actions as velocity operators within a CANN-inspired latent manifold provides a semantically meaningful alternative to tokenized or discrete latent actions (as in VQ-VAE methods).

(+) The system successfully transfers learned latent actions to unseen contexts: including cross-object, cross-domain, and cross-dataset generalization. Performance on OOD datasets like COIL-100 and Franka Kitchen is notably strong relative to prior work.

(+) The authors conduct extensive analyses (e.g., UMAP visualization, in-class structural sharing, cosine similarity of transitions) that clarify how HPC and MEC embeddings differ in representational function.

**Weaknesses:**

(-) Evaluation remains perception-oriented. While the model demonstrates visual prediction and generalization, it has not yet been tested in interactive or control settings where learned latent actions must guide decision-making or robotic policies. The connection to actual action planning thus remains conceptual rather than empirical.

(-) Limited ablation on factorization design. The paper assumes the separation between content (HPC) and structure (MEC) is beneficial but provides only one ablation with a unified latent space. Addition studies, such as varying the coupling strength or testing partial disentanglement, would strengthen the argument for this design.

(-) Biological analogy not fully justified. While the HPC–MEC analogy is intellectually appealing, the correspondence between biological circuits and deep modules is mostly metaphorical. The paper could temper claims about biological plausibility and focus more on empirical validation.

**Questions:**

How does the choice of latent action dimension and number of CANN modules affect generalization and reconstruction quality?

Can the proposed framework be extended to goal-conditioned prediction or hierarchical planning (e.g., using latent actions as building blocks for policies)?

Does the system exhibit any drift or instability when performing long-horizon autoregressive predictions without visual feedback?

Have the authors compared training with and without the fixed pretrained VQ-VAE encoder to assess the benefit of freezing visual features?

Could the learned latent actions be composed or interpolated to form new dynamics (e.g., blending rotation and translation)?

---

> ### Author Response · Authors · 2025-11-19
> **Rebuttal by authors (Part 1/2)**
>
> Thank you for your thoughtful review and insightful comments. We sincerely thank you for your positive feedback on our primary contributions. We are also encouraged by your appreciation for our empirical results as a key strength, including the model's strong OOD generalization and the extensive analyses.
>
> **Weakness 1: The connection to action planning**
>
> Thank you for this valuable point. You are correct that the current evaluation is perception-oriented. The core contribution of this work is to learn a disentangled world model and abstract latent actions, and to elucidate the functional correspondence between this model and the brain's HPC-MEC circuit. Visual prediction and zero-shot action transfer are fundamental tasks for validating the **disentanglement** and **abstraction** capabilities of these representations. We also focus on analyzing the disentangled representations to interpret the learn latents.
>
> We strongly agree that evaluating our HPC-MEC world model on robotic tasks is an important direction. A key part of this is accurately decoding latent actions into ground-truth actions. To measure this decoding ability directly, we supplemented our paper with an experiment that classifies action types from the learned latent states. We constructed a new dataset using OmniObject3D with sequences containing different transformations (rotation, horizontal/vertical translation, and scaling). Each sequence features an object where the category, initial position, orientation, and size are randomized. We fed these sequences to our model for zero-shot generalization and used the resulting latents to decode the **action type**.
>
> Table A
>
> | Embedding | $ p^{inf} $ | $ g^{inf} $ | $ \Delta p^{inf} $ | $ \Delta g^{inf} $ | $ z $ |
> | --- | --- | --- | --- | --- | --- |
> | Action Decoding Accuracy↑ | $0.3330 \pm 0.0163$ | $0.3486 \pm 0.0156$ | $0.8386 \pm 0.0263$ | $0.8868 \pm 0.0212$ | $\mathbf{0.9064 \pm 0.0145}$ |
>
> We found that our latent action z decodes the action label with the highest accuracy, followed by the MEC embedding change ($ \Delta g$). This quantitatively demonstrates that our model effectively disentangles action-specific information into these latents. That said, action planning is a critical next step. Our follow-up work is focused on implementing latent Model Predictive Control (MPC) on top of this HPC-MEC world model to accelerate the sampling and convergence process, as detailed further in our response to your **Question 2**.
>
> **Weakness 2: Limited ablation on factorization design**
>
> Thank you for this suggestion. If we understand correctly, the **"coupling strength"** you refer to would be the connection weights between the HPC and MEC modules. In our design, these are not manually controlled hyperparameters but are learned during training.
>
> Regarding **"testing partial disentanglement,"** we believe this can be achieved by controlling the strength of the bottleneck. We can fix the HPC (content) dimension while varying the MEC (structure) dimension, creating a transition from a unified model to our current, heavily bottlenecked setting. Our experience and sensitivity analysis (Appendix A.5) align with this intuition. We found that as the MEC dimension _increases_ (i.e., the bottleneck weakens), the latent action $ z $ becomes entangled with this excess content information. This directly results in the "texture leakage" from the source video during the reuse task, as seen in our "unified latent space" ablation (Table 2).
>
> **Weakness 3: Temper claims about biological plausibility**
>
> Thank you for this feedback, and we will revise the manuscript to temper our claims accordingly. To clarify, our model is best described as **bio-inspired** rather than bio-plausible. Our primary intent was to establish a functional correspondence to motivate future work, not to assert biological fidelity. We will make this "bio-inspired", functionalist approach much more explicit in the revision.
>
> Here, we included the neuroscience background not merely as a metaphor, but to ground this inspired design in empirical validation. Specifically:
>
> 1. The core HPC-MEC function of separating content (HPC) and structure (MEC) is what empirically enables our model's zero-shot generalization (Table 1) and interpretable latent space (Fig. 5).
> 2. The inductive bias from the CANN module was also empirically tested and found to be more effective for this task than a standard MLP (Table 2), validating its function.
>
> We sincerely appreciate your suggestion and will revise the text to focus more precisely on these functional justifications.

---

> ### Author Response · Authors · 2025-11-19
> **Rebuttal by authors (Part 2/2)**
>
> **Question 1: Sensitivity of latent dimension**
>
> For the **number of CANN modules** (MEC dimension), our current model uses 4096 modules. Reducing this to 2048 still allows the model to encode scene-specific dynamics, but with a noticeable loss of detail and increased blurriness in the generated images. A further reduction to 1024 exacerbates this issue and leads to convergence difficulties during training. Regarding the **latent action dimension**, we currently use a dimension of 2048. We found that the model can still predict the next frame with a dimension of 1024, though the generation quality is compromised. Compressing the latent action dimension further makes convergence very difficult, often causing the model to learn a trivial solution where it simply outputs the previous frame as its prediction.
>
> **Question 2: Goal-conditioned planning**
>
> Yes, absolutely. This is a key direction for our follow-up work, where we are applying this framework to goal-conditioned planning using Model Predictive Control (MPC). Our hierarchical HPC-MEC model is particularly well-suited for this. Beyond just separating content and structure, it naturally enables a separation of **stable** and **plannable** representations:
>
> + **Stable representations (HPC)** tell the agent "where I am" by encoding content-specific, contextual details.
> + **Plannable representations (MEC)** tell the agent "which direction I should go" in the latent space, acting like a "latent compass".
>
> Because the MEC space isolates abstract structure (akin to pose information), a direction calculated in this space is more robust and content-invariant. We can then **accelerate the sampling process of MPC** by using this directional information from the MEC to guide the search. The learned latent actions are then used to perform the "imagination" (rollouts) within our world model to find the optimal plan.
>
> **Question 3: Long-horizon prediction instability**
>
> Yes, the system does exhibit drift when performing long-horizon predictions. While all autoregressive models suffer from compounding error, we would like to point to **Table 1**, which shows our model **quantitatively outperforms all baselines** (LAPA, Moto, AdaWorld) in the autoregressive setting, achieving a significantly lower (better) LPIPS score.
>
> Mitigating this compounding error was not the primary scope of our current work, as this behavior is consistent with the biological system we are inspired by: the MEC's grid cells also accumulate errors during path integration.
>
> However, we are actively exploring this in our future work by modeling latent actions using **Lie group operators**. Our preliminary findings suggest that the structural properties inherent to Lie groups (such as element invertibility and operator orthogonality) can significantly mitigate this compounding error. This is a particularly promising direction because rotation operators are mathematically equivalent to CANN dynamics. This equivalence means we could integrate Lie operators directly into our current framework to overcome autoregressive error accumulation, an improvement we hope to demonstrate in a subsequent paper.
>
> **Question 4: Ablation on pretrained VQ-VAE**
>
> To answer your question directly: training "without a fixed VQ-VAE" would require a pixel-level reconstruction loss. We found that this would require us to increase our HPC, MEC, and latent action dimensions by at least 16-fold just to match the original model's generation quality. This would be counter-productive, as it would destroy the low-dimensional structure and abstraction of the latent actions.
>
> The intuition behind this result is that latent action models often face a **trade-off where pixel-level reconstruction quality conflicts with latent action abstraction**[1]. Models like LAPA, Moto, and AdaWorld exemplify this, often sacrificing generation quality for better abstraction. This suggests that a pixel-level loss is suboptimal for this task. Our strategy was to use the pretrained VQ-VAE's compressed latent embedding as the prediction target, which eases the learning burden.
>
> [1] AdaWorld: Learning Adaptable World Models with Latent Actions, ICML 2025
>
> **Question 5: Latent actions compositionality**
>
> That is a very interesting question! We found that composing translations (e.g., horizontal + vertical) **is linear**. As shown in Table 6 and Figure 12, simply adding their latent actions works correctly. However, we found that composing rotation and translation is _not_ linear in our latent space. Our initial attempts to simply add these latent actions did not produce the correct combined transformation in the image. We believe the composition of rotation/scaling with translation is a **non-linear operation**. While we have not yet identified the exact mathematical form, our future work is exploring the use of **Lie group operators** to provide the correct mathematical language for this complex, non-linear composition.

---

### Author Response · Authors · 2025-12-03
**Summary of Rebuttal**

Dear Area Chairs and Reviewers:

We sincerely thank the reviewers for their insightful feedback and the Area Chairs for their time and effort in assessing our work. To assist in your final evaluation, we summarize below the review progress, highlighting the strengths recognized by the reviewers and our responses to key concerns.

The initial ratings are **8, 8, 4, 2**. **Before the OpenReview incident, Reviewer kySr maintained a score of 8, and we believe we have addressed the primary concerns raised by Reviewer Fvm6.**

**Brief Summary of Our Work:** We propose a **brain-inspired**, self-supervised framework that learns latent actions and a disentangled world model solely from real-world videos. Our model introduces disentanglement of content (HPC) and structure (MEC), and models latent actions as velocity operators within a CANN-inspired latent manifold. In doing so, we highlight the **largely underexplored** functional correspondence between our model and the brain's HPC-MEC circuit.

## Reviewer Consensus on Strengths

- **Biological Inspiration & Architecture Design**:
  Reviewers praise the "elegant synthesis between computational neuroscience and self-supervised world modeling" (Reviewer fVRp), noting the HPC–MEC–inspired design as "strong novelty" and "well-justified" (Reviewer kySr), and likewise appreciating its biologically grounded motivation (Reviewer Fvm6). Modeling latent actions as CANN-based velocity operators is seen as a "semantically meaningful alternative" (Reviewer fVRp).
- **Evaluation & Generalization**:
  The model shows "strong autoregressive performance" (Reviewer T6rd), learns visually independent latent actions (Reviewer T6rd, Fvm6), and transfers effectively across content (Reviewer fVRp, Fvm6). Its OOD generalization performance is repeatedly highlighted (Reviewer fVRp, kySr, Fvm6).
- **Analysis & Interpretability:** Reviewers highlight the "extensive analyses" (Reviewer fVRp) and "qualitative evidence" (Reviewer Fvm6) that confirm the separation of HPC and MEC. They value how these visualizations "clarify how HPC and MEC embeddings differ" (Reviewer fVRp) and note that the ablations demonstrate the "relevance of the used predictive CANN model" (Reviewer T6rd).
- **Clarity & Reproducibility**:
  Reviewers find the presentation clear (Reviewer kySr) and the code well-structured and easy to use (Reviewer T6rd).

## Response to Reviewer Concern

**Reviewer Fvm6 (Score 2): VQ-VAE ablations & Action decoding**

- **Primary Concern**: The lack of VQ-VAE ablations and the absence of quantitative metrics for action decoding.
- **Further Comments**: The reviewer **has acknowledged that the several main concerns are addressed** but requested further visualizations and results against baselines for action decoding.
- **Our Rebuttal:**
  - **New Ablation on VQ-VAE**: We add the requested ablation studies to isolate the source of our performance gains.
  - **New Action Decoding Results**:  We add new quantitative results showing that our model outperforms all other baselines.
  - **Additional Visualization of All Baselines**:  We add visualizations in **Appendix F.3**. The results show that the baselines struggle with effective action transfer.

**Reviewer fVRp (Score 4): Planning Task & Factorization Ablations**

- **Primary Concern:** Lack of evaluation in interactive control settings and requested ablations to justify the factorization design.
- **Our Rebuttal:**
  - **Clarification on Control Scope:** We introduce action decoding accuracy as a quantitative proxy for planning capability. Our model outperforms all other baselines and ablations on this metric. We also detail how the HPC-MEC separation naturally facilitates Model Predictive Control (MPC).
  - **Factorization Justification:** We emphasize our sensitivity analyses in **Appendix B.5**, demonstrating that looser bottlenecks lead to representation entanglement.

**Reviewer T6rd (Score 8): Clarifications**

- **Primary Concern:** The clarifications for the HPC-MEC functional separation and the details of specific ablation models.
- **Our Rebuttal:**
  - **Architectural Clarification:** We clarified the role of the MEC module as the essential information bottleneck.
  - **Ablation Justification:** We detailed our descriptions of our ablation models.

**Reviewer kySr (Score 8): Robotic application**

- **Primary Concern:** The absence of downstream robotic tasks.
- **Our Rebuttal:**
  - **Downstream Utility:** We validated the model's practical utility through a new action decoding experiment as a quantitative proxy for control tasks.
- **Discussion Status:** The reviewer has maintained a score of 8, reaffirming the assessment that the paper merits this high rating.

We hope this summary provides a clear overview of our model's contribution and the progress made during the rebuttal period. Thank you again for your time and for managing this assignment under the challenging conditions.

Best Regards,

The Authors

---

### Meta-Review · Area_Chair_RuWf · 2025-12-08

**Summary:**

This paper proposes a hippocampal–entorhinal–inspired latent world model with two hierarchical latents: a high-dimensional “HPC” latent for content and a lower-dimensional “MEC” latent for structure, on which a CANN-style residual dynamics is applied. Latent actions are learned by an inverse model from consecutive MEC latents and reused across objects and datasets for video prediction and action transfer. While the experiments show some interesting qualitative phenomena (e.g., cross-object 3D rotation, OOD action transfer, representation analyses of HPC vs MEC), my main concern is that the methodological novelty is almost entirely concentrated in the HPC/MEC separation, and in implementation this separation reduces to stacking two spatiotemporal Transformer layers with a dimensionality bottleneck on the last layer and attaching dynamics and losses only there. The work also does not provide strong evidence that the proposed architecture is essential relative to existing latent-action world models, and its evaluation does not match the strength of its claims about “abstract latent actions for control”. Overall I lean to rejection.

**Reviewer Concerns:**

1. Methodological novelty and relation to prior work.
  I largely agree with the critical reviewer that, from a “latent action + world model” perspective, most ingredients (VQ-VAE backbone, inverse-dynamics latent actions from frame/latent differences, residual latent dynamics, action transfer) have appeared in prior work such as LAPA, AdaWorld, Moto, etc. The main architectural distinction here is the explicit HPC/MEC decomposition: a high-dimensional latent followed by a lower-dimensional latent onto which all dynamics and action losses are attached. In practice this is implemented as two serial spatiotemporal Transformer blocks with a dimensionality bottleneck on the second block. The paper does not include ablations that remove the HPC-only reconstruction head or collapse the two blocks into a unified latent with matched capacity, so it is difficult to assess whether the explicit HPC/MEC separation provides a principled, indispensable innovation rather than a relatively minor re-parameterization of a large Transformer with a bottlenecked final layer. The rebuttal adds some VQ-VAE ablations and an action-decoding metric, but these do not fully resolve this attribution issue.

2. “Abstract latent actions” and evaluation gap.
  The paper makes strong claims about learning abstract, reusable latent actions in a hippocampal–entorhinal-inspired world model and repeatedly frames the work as relevant to control and planning. However, the training objective only guarantees that the continuous latent (z_t) encodes MEC latent differences and supports one-step/short-horizon prediction; there is no variational bottleneck or causal constraint that would enforce environment-agnostic “action factors” in a strong sense. The added action-decoding results in the rebuttal show that (z_t) indeed contains motion information, but they are limited in scope and do not fully answer the concern that (z_t) may function mainly as a compressed code for (\Delta g), rather than a clearly disentangled, abstract action representation. Moreover, the evaluation is almost entirely in terms of video prediction and qualitative action transfer; there is no systematic comparison on standard downstream control or planning benchmarks, nor a direct head-to-head against LAPA/AdaWorld/Moto under shared backbones. This makes the “world model for control / abstract latent actions” narrative feel oversold relative to the experiments.

3. Positive aspects and what the rebuttal did address.
  On the positive side, the paper does present non-trivial representation analyses comparing HPC vs MEC, especially on 3D rotation / OOD datasets, and demonstrates that a MEC-based latent (plus dynamics) can yield more shared trajectories and better cross-object transfer than a unified latent baseline (which tends to exhibit texture leakage). These results suggest that the chosen hierarchical design has some empirical merit as a representation for “abstract structure”. The rebuttal also adds a VQ-VAE ablation and baseline visualizations, which alleviate (but do not fully remove) the concern that the gains are purely due to a stronger visual backbone. Still, the core questions about unique methodological contribution and the strength of the “abstract action for control” claim remain largely outstanding.

Given the above, I do not think the rebuttal sufficiently addresses the central concerns about novelty attribution and evaluation mismatch, even though it does patch some peripheral issues.

**Reviewer Scores:**

Reviewer T6rd (score 8, positive):
This reviewer focused on the clarity of the architecture and the empirical performance on video prediction and transfer. After discussion, I believe they might slightly temper their enthusiasm in light of the limited methodological novelty and the missing control benchmarks, but would likely remain positive overall (e.g., 7–8). I acknowledge their perspective that the engineering and representation-analysis aspects are interesting, but I weigh novelty and evaluation alignment more heavily.

Reviewer kySr (score 8, positive):
Similar to T6rd, kySr appreciated the neuro-inspired story and broad set of experiments. With the additional information from the rebuttal, they might keep a high score or mildly reduce it (7–8), seeing the work as a solid neuro-inspired representation paper. However, my assessment is that the evidence for a genuinely new architectural principle (beyond a bottlenecked Transformer) is not strong enough for this venue, so I diverge from their overall recommendation.

Reviewer fVRp (score 4, borderline/negative):
This reviewer raised concerns about the lack of downstream control tasks and limited factorization analysis. The rebuttal does not add control experiments and only modestly strengthens the analysis of the action representation. I expect fVRp would likely keep a similar score (4–5). I broadly agree with their concerns and weight them significantly in my decision.

Reviewer Fvm6 (score 2, strongly negative):
The most critical reviewer emphasized unclear attribution of improvements to the new architecture vs. backbone, insufficient evidence for truly abstract actions, and the mismatch between the control-oriented narrative and the mostly predictive evaluation. The rebuttal adds a small VQ-VAE ablation, an action-decoding number, and more visualizations, but does not fundamentally change the picture. I suspect this reviewer would at best move slightly upward (e.g., 3–4) but remain negative. While I find some of their expectations quite strict, I agree with the core points about novelty and evaluation and, as a result, my final recommendation aligns closer to their stance: reject.

---

### Decision · Program_Chairs · 2026-01-26

Reject